**communications** engineering

# Development of open radio access networks (O-RAN) for real-time robotic teleoperation
Saber Hassouna [1], Jaspreet Kaur[1], Burak Kizilkaya[1], Jalil ur Rehman Kazim[1], Shuja Ansari[1], Arzad Alam Kherani[2], Brijesh Lall[3], Qammer H. Abbasi [4,5] ✉ & Muhammad Imran [1]

Open Radio Access Networks (O-RAN) offer a flexible RAN architecture for future 6G systems, yet their complexity and lack of real-world testbeds pose interoperability challenges, particularly with emerging software platforms and robotic systems. Here we present a real-world software-defined radio testbed based on an open-source 4G long-term evolution (LTE) system, integrated with the near-real-time (Near-RT) RAN Intelligent Controller (RIC) via standard O-RAN E2 interfaces. It enables connectivity with robotic end devices such as a haptic controller and robotic arm, demonstrating the activation of E2 functionality within a live RAN environment. The testbed enables haptic operation with sub-one-second latency and block error rate (BLER) under 12% for tasks such as dental inspection use cases. We also demonstrate replacement of software-defined radios (SDRs) with low-power mobile dongles, achieving comparable 10 Mbps throughput while cutting power consumption by 90%. This setup establishes a foundation for advancing research and integration in managing next-generation RANs.

The fifth generation (5G) of mobile communication has been driven by the need for high data rates, energy-efficient communication, and seamless connectivity. These requirements have led to the classification of 5G services into three main categories: enhanced mobile broadband, ultra-reliable low-latency communication (URLLC), and massive machine-type communication (mMTC). This categorization positions 5G as a key enabler for various applications, particularly in the industrial internet of things (IIoT)[1], as well as in healthcare[2] and education[3], where many use cases rely on the URLLC framework.

Interest in open-source platforms for cellular networks has surged in recent years, with tools like software radio system radio access network (srsRAN)[4] and open air interface (OAI)[5] providing researchers accessible alternatives to traditional networks. Paired with commercial software-defined radios (SDRs), these platforms enable quick setup of standard-compliant network components compatible with commercial devices[6]. The rise of open radio access network (O-RAN) has further simplified development, fostering real-world testing and innovation aligned with industry standards. Emerging applications like augmented reality (AR), virtual reality (VR), ultra high definition (ultra-HD) streaming, autonomous vehicles, and IIoT require wireless systems with high throughput, low latency, and exceptional reliability. To meet these demands, innovative and programmable radio access network (RAN) architectures have been developed, with O-RAN widely adopted.

Spearheaded by the O-RAN alliance, this architecture promotes open, flexible, interoperable, and intelligent networks. By enabling disaggregated and adaptable designs, O-RAN allows operators to build cost-effective networks that support the growing demands of next-generation applications.

The O-RAN architecture introduces a structured approach to RAN resource control, managed by high-level orchestration and automation components such as policies, configurations, and the non-realtime RAN Intelligent Controller (RIC)[7] which supports third-party microservices called rApps. These components interact with the near-realtime RIC[8] through the A1 interface. The near-realtime RIC oversees RAN nodes (e.g., eNodeB/gNodeB, radio unit (RU)/distributed unit (DU)) and extends its functionality through custom third-party xApps. These xApps, deployed as low-latency cloud services, connect to RAN nodes and the RIC via the E2 interface, enabling enhanced flexibility and control[9,10].

Implementing intelligent, data-driven, programmable, and virtualized networks presents several challenges. Greater clarity is needed on the open-source frameworks designed to support research efforts. For instance, while the O-RAN Software Community (OSC)[11] provides the codebase for implementing O-RAN architecture and interfaces, it lacks direct integration with existing RAN implementations or the RAN itself. Additionally, the scarcity of comprehensive use cases and systematic implementation processes hampers progress. Other hurdles include unclear functionalities and parameters for

[1]University of Glasgow, Glasgow, United Kingdom. [2]Indian Institute of Technology Bhilai, Raipur, Chhattisgarh, India. [3]Indian Institute of Technology Delhi, Delhi, India. [4]James Watt School of Engineering, University of Glasgow, Glasgow, UK. [5]Distinguished Visiting Research Fellow, Abu Dhabi University's College of Engineering, Abu Dhabi, UAE. ✉e-mail: qammer.abbasi@glasgow.ac.uk

each network element and the underexplored potential of artificial intelligence (AI) and data analytics in optimizing wireless network performance[12].

Recent efforts have sought to address these challenges and foster knowledge-sharing within the O-RAN research community. The authors[13] introduced AI controllers integrated into the O-RAN architecture. The researchers[14] detailed the software implementation of O-RAN architecture. The study[15] surveyed key O-RAN technologies, including disaggregation, virtualization, and slicing. The reference[16] presented O-RAN tools alongside Colosseum and an SDR-based wireless channel emulator with long-term evolution (LTE) capabilities. Additionally,[10] demonstrated a closed-loop integration of O-RAN-compliant software for data collection on the platform for open wireless data-driven experimental research (POWDER) testbed, featuring an open-source softwarized cellular network. While these testbeds support 4G experiments and open-source solutions, they lack comprehensive designs and integration details for building a complete end-to-end O-RAN research platform. The efforts[12] introduce the components of an O-RAN research testbed built entirely with commercial off-the-shelf hardware and open-source software. It evaluates hardware design options and provides a detailed overview of software suites, focusing on the near-real-time RAN Intelligent Controller (near-RT RIC) deployment and E2 interface implementation.

In this paper, we have developed a fully open-source O-RAN testbed platform using SDRs. This testbed integrates an LTE network with a near-real-time RIC, enabling seamless interaction with hosted xApps via open interfaces. We utilize the srsRAN[17] 4G software suite, an open-source cellular solution, to build our LTE network. This suite includes key components such as an LTE core, a 4G base station, and user equipment (UE) to emulate a fully functional LTE network. We model the LTE network with two UEs for a robotic dental inspection use case, where one UE functions as a srsUE haptic controller and the other as a srsUE robotic arm. An O-RAN-enabled teleoperation testbed is developed to evaluate communication latency and packet loss, showcasing the potential of O-RAN networks for real-time remote applications like robotics in healthcare.

Our primary objective was to demonstrate the feasibility of deploying an open, standards-aligned, end-to-end O-RAN system using open-source platforms (srsRAN and OpenAirInterface components) for real-time robotic teleoperation. While it is true that the use of 5G would provide lower latency and higher reliability, our motivation for selecting a 4G LTE stack was grounded in practical accessibility and current deployment maturity.

We conducted all experiments using srsRAN-LTE v21.10, combined with the openAI cellular (OAIC) E2 agent and the Cherry v2.0 Near-RT RIC container. This version was selected because it represented the most recent stable LTE release available when data collection began (May 2025), and it supported a robust 1-s E2SM-KPM telemetry stream, critical for enabling our sub-second control loop in robotic teleoperation.

We evaluated srsRAN-5G v24.04, but encountered several blockers that prevented reliable experimentation: (1) the open-source branch at the time supported only E2 service model-key performance measurement (E2SM-KPM) and a single radio control (RC) Style 2, without O1 interface support; (2) community issue[18] #1227 (UE deregistration in standalone (SA) mode) consistently reproduced in our lab; and (3) the SA UE exhibited missing counters and intermittent segmentation faults, which disrupted closed-loop telemetry collection.

Given these issues, we chose to proceed with the mature LTE stack for this study. However, we note that srsRAN-5G v24.10, released after our experiments, has introduced additional features, including central unit (CU), DU split, E2SM-RC Style 3, and stability improvements. Once support for SA handover, wider bandwidths (e.g., 30 kHz subcarrier spacing), and a complete O1 interface is fully stabilized in the open branch, our modular testbed can be readily migrated to a native 5G stack without architectural changes. Our software design already anticipates this forward compatibility.

The open-source srsRAN 5G stack[19], while evolving, currently exhibits limitations in stability and full-stack integration, particularly at the UE and RIC interface layers, required for consistent and reproducible teleoperation testing under tight latency constraints. As per the srsRAN documentation[20], the E2 interface is still under development with limited features, and the

srsUE component has constraints in 5G SA mode, including limited bandwidth support and lack of handover functionality[21]. Additionally, stability issues[18] such as UE disconnections after a few minutes have been observed[22,23].

We emphasize the following to show the specific contributions of our O-RAN integration:

- The system incorporates full integration of a Near-RT RIC with srsRAN using E2 interfaces and two functional xApps, an extended version of the key performance indicator monitor (KPIMON) xApp and a custom xApp, within a robotic teleoperation context. Our architecture preserves the monolithic nature of the 4G eNB, with no additional separation beyond what is inherently present in srsLTE's software design. This integration, combining Near-RT RIC, E2 interfaces, and robotic teleoperation, has been less explored in existing literature.
- The extended KPIMON xApp, based on the OAIC implementation[24], was used for real-time sensing of network telemetry and RIC indication messages. Specifically, the KPIMON SM was configured to periodically monitor the packet data (PDCP) bytes transmitted to and from the srsUE haptic interface and the robotic arm, with a 1-s event trigger interval, while both UEs maintained continuous connectivity to the eNB. This xApp-enabled network state awareness at the RIC level without requiring deep packet inspection.
- The custom xApp, also based on OAIC templates[25], was designed to monitor modulation and coding scheme (MCS) alongside block error rate (BLER) using a lightweight machine learning algorithm. The simple rule-based classification is used to detect degradation patterns in MCS/BLER trends without invoking complex or computationally heavy inference models, as long as BLER and latency remain within defined tolerable limits. The purpose of this monitoring framework is to establish the groundwork for future closed-loop control extensions, including dynamic MCS reconfiguration, predictive congestion management, or adaptive scheduling, all of which can be built incrementally atop the current xApp framework.
- Overall, this setup introduces an application-aware O-RAN telemetry pipeline, where robotic quality of service (QoS) constraints (e.g., low BLER and sub-50 ms latency) are linked to RAN behavior, a first step toward cross-layer control in open RAN-enabled cyber-physical systems. This platform can serve as a stepping stone for future 5G-based robotic and XR applications

## Methods

This section outlines the architectural design and implementation details of the O-RAN testbed, as well as the data transmission and control mechanisms underpinning the real-time robotic teleoperation.

### O-RAN network design and implementation

The proposed system architecture is shown in Fig. 1 of the manuscript, which provides a high-level overview of the Open RAN-based framework integrating robotic control and haptic feedback via a fully deployed 4G LTE testbed. This includes the near-real-time RIC, the custom xApp, and the end-to-end connectivity between the operator-side haptic interface and the teleoperator robotic arm. Figure 2 further clarifies the experimental testbed setup, including the servers, universal software radio peripherals (USRPs), mobile dongles, and UE hardware. Figure 3 illustrates the physical deployment of the base station setup and connected devices.

The system transmits real-time control commands, such as position and velocity vectors from the haptic interface to the robot, and force feedback signals in the reverse direction. These are encapsulated in standard IP packets over LTE and routed through the srsRAN-based evolved packet core (EPC). In addition, the iperf3 was used to generate UDP traffic in some test phases to characterize the LTE throughput under controlled load conditions, as described in the OAI Cellular documentation[26].

### O-RAN system for real-time robotics. 
O-RAN represents an important advancement in the development of next-generation cellular networks,

**Fig. 1 | Open RAN (O-RAN) system integrating robotic teleoperation and haptic control, with functional split Options 8 and 7.2 in the uplink and downlink.** The architecture includes the near-Real-Time RAN Intelligent Controller (near-RT RIC), Open Central Unit (O-CU), Open Distributed Unit (O-DU), and Open Radio Unit (O-RU), supporting real-time data flow between the haptic controller and robotic arm.

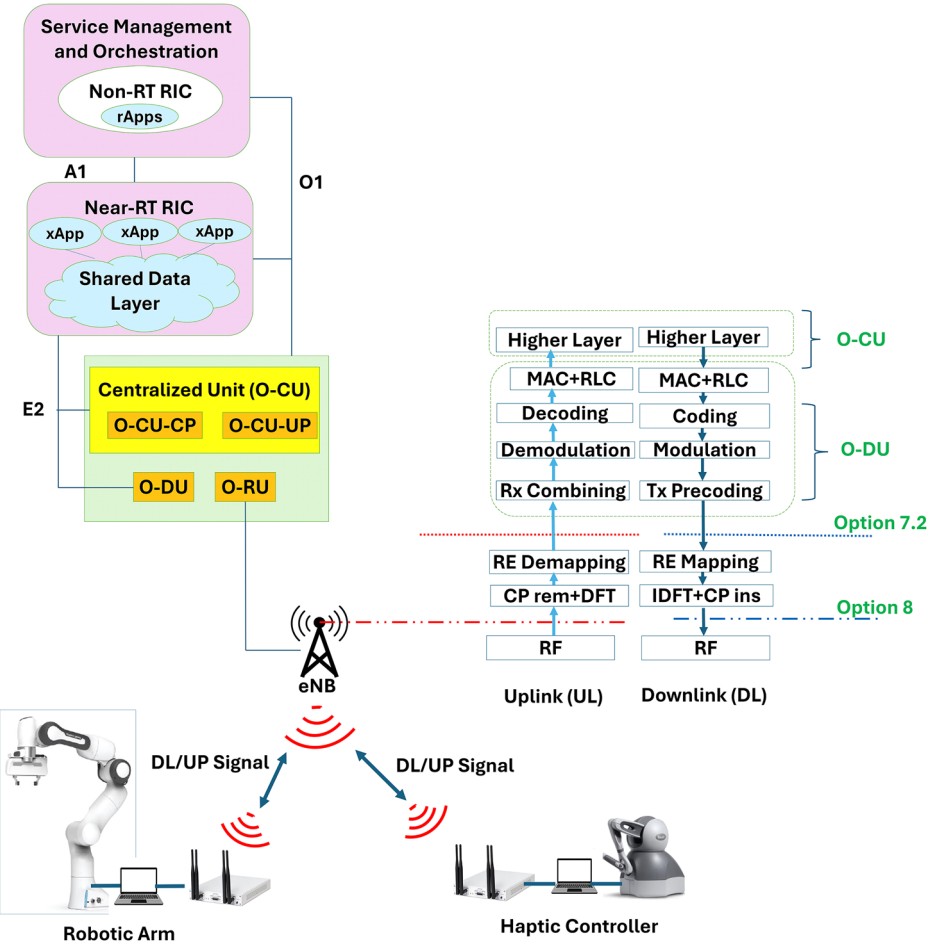

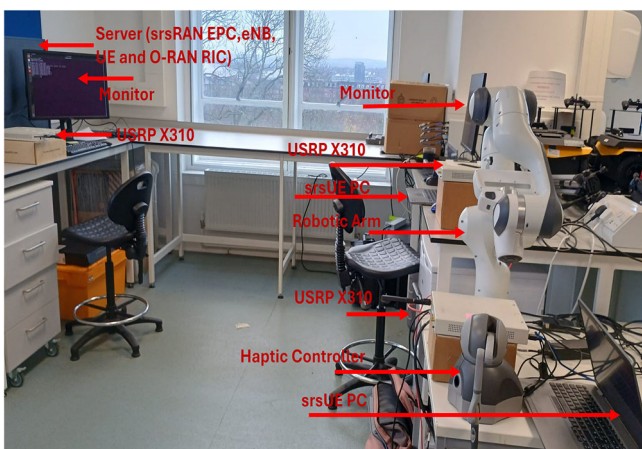

**Fig. 2 | Experimental Open RAN (O-RAN) testbed for robotic teleoperation.** The testbed integrates the Software Radio Systems Radio Access Network (srsRAN) Evolved Packet Core (EPC), the Evolved Node B (eNB), and the RAN Intelligent Controller (RIC) with a haptic controller and a robotic arm. This configuration enables real-time end-to-end control of the robotic arm through the O-RAN-based LTE network.

offering open interfaces and integrating AI to enhance the deployment, operation, and management of RAN. In this work, we concentrate on key components within the O-RAN architecture illustrated in Fig. 1 including the robotics. The RIC is a crucial component of O-RAN, functioning as a software-defined subsystem that optimizes RAN operations and manages various units in the open access network

through standardized interfaces[15]. Two types of RICs, the non-RT RIC and near-RT RIC, enable network function disaggregation and ensure seamless multi-vendor interoperability, allowing third-party vendors to develop microservice-based applications known as xApps and rApps. These applications, deployed in the near-RT RIC and non-RT RIC, respectively, automate and enhance RAN operations. The near-RT RIC is positioned at the network edge and connects to the CU and DU via the E2 interface, as illustrated in Fig. 1. Operating within a time frame of 10 ms to 1 s, it includes multiple microservice applications called xApps, a shared data layer (SDL) containing RAN information, and messaging infrastructure for communication with system components. The near-RT RIC interacts with the non-RT RIC through the A1 and O1 interfaces, leveraging real-time RAN data to optimize and manage tasks across the network. An xApp is a plug-and-play, microservice-based software application deployed in the near-RT RIC to perform specific functions or services. These xApps, often developed by third parties, utilize the data stored in SDL for tasks such as resource management, RAN data analysis, and RAN control. After processing the data, xApps relay control decisions back to the RAN via the E2 interface.

In this work, two open-source xApps from the OAIC project were integrated into our testbed and deployed in the Near-RT RIC environment:

- KPIMON xApp[27] used to monitor real-time network statistics such as PDCP bytes, channel quality indicator (CQI) reports, and UE identifiers via E2SM-KPM service model. It was configured with a 1-s periodic trigger to log performance data for both the haptic controller and the robotic arm UEs.

- Custom xApp[25] used to monitor and visualize MCS and BLER trends over time. It runs a lightweight Python process that subscribes to telemetry and displays variations in link-layer parameters.

Importantly, no modifications were made to either xApp. The MCS selection mechanism within the srsRAN eNB was left unmodified, adhering to standard 3GPP procedures where MCS is selected based on CQI feedback derived from signal-to-interference-plus-noise ratio (SINR) and hybrid automatic repeat request (HARQ) performance. The custom xApp does not override or inject control commands into the RAN, nor does it interfere with the eNB's decision-making logic. Consequently, the xApp in this work is monitoring only (no override of 3GPP MCS logic); it processes 1s E2SM KPM indications in the Near RT RIC, while policy-level actions via A1/rApps are left to future extensions.

The primary role of these xApps in our study was to establish a working telemetry and analytics pipeline, enabling real-time monitoring of RAN behavior in a robotic teleoperation scenario. This architecture serves as a foundation for future closed-loop control, where KPIMON data could be used to trigger xApp-driven optimization policies (e.g., adaptive MCS tuning or latency-aware scheduling).

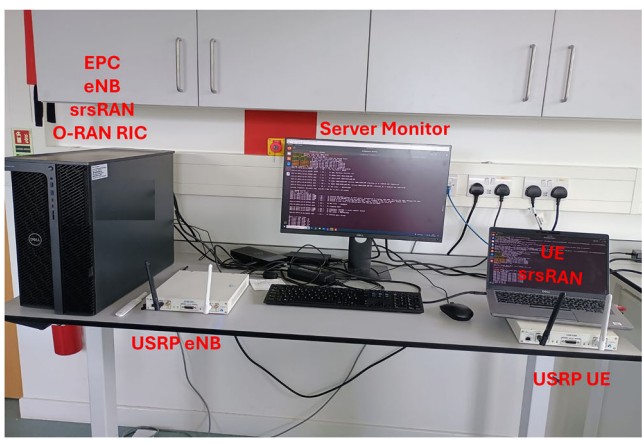

**(a)**

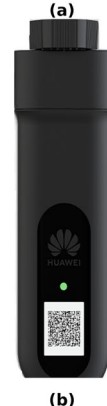

**(b)**

**Fig. 3 | Experimental setup showing the O-RAN base station and user equipment. a** Base station and server setup including the Evolved Packet Core (EPC), eNodeB (eNB), and Radio Intelligent Controller (RIC) hosted on a server, connected to USRP devices. **b** Huawei E3372 mobile dongle used as user equipment (UE).

Despite not implementing closed-loop control, the system achieved acceptable real-time performance, maintaining downlink latency below 50 ms and BLER within tolerable limits, thus validating the feasibility of using lightweight, standards-aligned xApps for robotic use cases under O-RAN architecture.

The O-RAN system in Fig. 1 featuring haptic feedback, tailored for emerging internet of skills use cases. A 3D System haptic device utilized as the haptic controller, offering six degrees of freedom (6-DoF) for positional sensing and three degrees of freedom (3-DoF) for force feedback. Using this device, the human operator manipulates a remote robotic arm to achieve a task via the O-RAN network. We utilized the Franka Emika Panda robotic arm, a 7-DoF serial manipulator with a control and sensor sampling rate of 1 kHz. In the healthcare application, we explored a use case of remote dental inspections, enabling skilled dentists to perform these procedures with enhanced reliability and latency within acceptable limits[28].

**O-RAN prototype with haptic controller and robotic arm.** As illustrated in Fig. 2, the srsEPC core and srsRAN eNB are deployed on the same server, which interfaces with a USRP X310 SDR for over-the-air communication. Two additional workstations host srsRAN UEs, each configured to connect to the LTE network through separate USRP X310 SDRs for wireless communication. The downlink (DL) and uplink (UL) setting frequencies for the LTE network are 2680.0 MHz and 2560.0 MHz, respectively. The first workstation's srsRAN UE is integrated with a haptic controller, while the second workstation's srsRAN UE is linked to the robotic arm. Both the haptic controller and the robotic arm utilize their respective USRP X310 SDRs to connect to the O-RAN base station (BS). Table 1 summarizes the testbed.

**O-RAN prototype to remotely control the robotic Arm using mobile dongles.** The prototype leverages the disaggregated architecture of O-RAN to enable real-time remote control of a robotic arm. A key component is the use of customized mobile dongles, specifically the Huawei EE32. The customization involved burning custom firmware onto the EE32 using the SysComms SIM Writer tool, integrating international mobile subscriber identity (IMSI) data from the User database stored in a comma-separated values file. This process enabled the dongle to register as a UE device, successfully generating traffic and providing internet connectivity through the O-RAN basestation. Notably, the mobile dongle enhances the system's mobility by replacing the SDR X300, offering a more compact and flexible solution. Furthermore, the dongle consumes only 4.5 W, compared to 45 W for the USRP, making it more power-efficient. Additionally, the dongle costs only a few dollars, whereas the USRP costs thousands, making it a much more affordable option. These enhanced features of the dongle over the USRP are vital for dynamic field operations and applications like dental inspection, where real-time remote control and low latency are critical. Figure 3 provides a magnified view of the base station and users depicted in Fig. 2. From Fig. 3, it is evident that the USRP can be replaced with the mobile dongle. However, we observe that both the dongle and the USRP demonstrate comparable bitrate performance. The performance comparison will be shown in the simulation results section.

**Data transmission model**

The dental inspection system under consideration is designed on the O-RAN architecture, as illustrated in Fig. 1. The setup includes a single base station featuring an open radio unit (O-RU) and K UEs (srsUE haptic

**Table 1 | The testbed features**

| Item | USRP | Dongle | CPU | RAM | NIC | MTU | RF channel | Teleoperation |
|------|------|--------|-----|-----|-----|-----|-----------|---------------|
| Server | X310 | / | 16 | 16 GB | 10 Gbps | 9000 Bytes | 2 × 2 | – |
| PC1 | X310 | Huawei EE32 | 8 | 8 GB | 1 Gbps | 1500 Bytes | 1 × 1 | Haptic controller |
| PC2 | X310 | Huawei EE32 | 8 | 8 GB | 1 Gbps | 1500 Bytes | 1 × 1 | Robotic arm |

controller and srsUE robotic arm) that are randomly distributed within the robotic lab. In this scenario, the robotic arm and the haptic controller are modeled as two UEs. Both the robotic arm and the haptic controller are equipped with a single antenna, while the O-RU is outfitted with $N$ antennas. The robotic arm and the haptic controller are positioned at Cartesian coordinates (1, 3, 0) and (0, 3, 0), respectively, relative to the O-RU as per Fig. 2. The O-RU connects to the bare-metal server deployment via fronthaul links, enabling virtualization and shared processing resources[29].

The bare-metal server deployment comprises two primary units: the O-CU and O-DU. As per the O-RAN specification, the O-DU handles lower-layer network functions (radio link control, medium access control (MAC), and physical layer (PHY)), while the O-CU manages higher-layer operations, as shown in Fig. 1. Additionally, O-RAN includes logical nodes: the near-RT RIC for near-real-time RAN resource optimization and the non-RT RIC, located within the Service Management and Orchestration (SMO) unit, for non-real-time orchestration. O-RAN supports various deployment options, including setups where the O-DU and O-CU are either co-located or geographically separated. This research adopts the scenario where the O-CU and O-DU are co-located to minimize the delay, with the near-RT RIC connected via the E2 interface as defined by O-RAN[30]. The O-RU supports the haptic controller and the robotic arm in a coherent manner to enhance throughput, spectral efficiency, and data rate. Let $x_{k,J} \in \{0, 1\}$ represent a binary variable indicating whether $UE_k$ is served by the O-RU. The variable is set to one if the O-RU serves $UE_k$, and zero otherwise. In our study, we have one O-RU in the O-RAN server which serves both UEs.

Let $\zeta_i \in \mathbb{C}$ represent the downlink data signal for $UE_i$, with $\mathbb{E}\{|\varsigma_i|^2\} = 1$. For $x_i = 1$, the precoding coefficient and the transmit power associated with $UE_i$ and O-RU are denoted as $\mathbf{w}_i \in \mathbb{C}^N$ and $p_i \geq 0$, respectively. In the physical testbed, the frequency-domain precoded signal to be transmitted by O-RU is formulated as:

$$\mathbf{x} = \sum_{i=1}^{K} \sqrt{p_i}\, x_i\, \mathbf{w}_i\, \varsigma_i \in \mathbb{C}^N. \tag{1}$$

The received frequency-domain downlink signal at $UE_k$ can be represented as follows:

$$y_k^{\mathrm{dl}} = h_k \mathbf{x} + n_k. \tag{2}$$

Substituting $\mathbf{x}$ from Eq. (1):

$$y_k^{\mathrm{dl}} = \sum_{i=1}^{K} \sqrt{p_i}\, x_i\, h_k\, \mathbf{w}_i\, \varsigma_i + n_k, \tag{3}$$

where $\mathbf{h}_k \in \mathbb{C}^N$ denote the frequency-domain channel from $UE_k$ to O-RU in an arbitrary coherence block. The channels are modeled as correlated Rayleigh fading, i.e., $\mathbf{h}_k \sim \mathcal{N}_{\mathbb{C}}(\mathbf{0}_N, \mathbf{R}_k)$, and are independent across different UEs and O-RUs. The correlation matrix $\mathbf{R}_k \in \mathbb{C}^{N \times N}$ reflects the spatial correlation of the channel $\mathbf{h}_k$ between the antennas of O-RU, with the average channel gain given by $\beta_k = \mathrm{tr}(\mathbf{R}_k)/N$[31]. The receiver noise is distributed as $n_k \sim \mathcal{N}_{\mathbb{C}}(0, \sigma^2)$. The downlink data rate of $UE_k$ at each resource block can be computed as follows:

$$R_k = B \log_2(1 + \mathrm{SINR}_k) \, \mathrm{bits\ s^{-1}}, \tag{4}$$

Where $B$ is the bandwidth (Hz) with effective SINR presented by:

$$\mathrm{SINR}_k = \frac{P_{\mathrm{sig}}}{P_{\mathrm{int}} + P_{\mathrm{noise}}}, \tag{5}$$

where, $P_{\mathrm{sig}}$ is the signal power for user, $P_{\mathrm{int}}$ is the interference power and $P_{\mathrm{noise}}$ is the noise power. The signal power for $UE_k$ is $p_k|h_k|^2|$, the interference power from other users UEs $i \neq k$ is $\sum_{i=1,i\neq k}^{K} p_i|h_k|^2|w_i|^2$ and the

**Table 2 | The number of PRBs allocated in bandwidth**

| Bandwidth [MHz] | 1.4 | 3 | 5 | 10 | 15 | 20 |
|---|---|---|---|---|---|---|
| Number of PRBs | 6 | 13 | 25 | 50 | 75 | 100 |

noise power is $\sigma^2$. The SINR becomes:

$$\mathrm{SINR}_k = \frac{p_k\, |h_k|^2\, \|\mathbf{w}_k\|^2}{\sum_{i=1,\, i\neq k}^{K} p_i\, |h_k|^2\, \|\mathbf{w}_i\|^2 + \sigma^2}. \tag{6}$$

The data rate can be finally given as:

$$R_k = B \log_2\left(1 + \frac{p_k|h_k|^2|\mathbf{w}_k|^2}{\sum_{i=1,\, i\neq k}^{K} p_i|h_k|^2|\mathbf{w}_i|^2 + \sigma^2}\right) \mathrm{bits\ s^{-1}}. \tag{7}$$

To provide a foundational understanding of how wireless channel conditions influence achievable data rates, we begin with a sequence of theoretical expressions, outlined in Eqs. (1) through (7). These equations illustrate the progressive refinement from idealized channel capacity models, based on Shannon's theory and SINR, to more practical representations that incorporate system-specific parameters such as modulation order ($Q_m$), coding rate ($R_c$), and allocated bandwidth in terms of Physical Resource Blocks (PRBs).

While these expressions are not directly used in the performance analysis, they serve to conceptually frame how radio conditions (e.g., SINR) are mapped into modulation and coding configurations, and ultimately, throughput. These intermediate steps also help contextualize the relationship between channel estimation quality and application-layer performance in our robotic teleoperation system.

Building upon this theoretical framework, we adopt the standardized PHY throughput expression defined in the O-RAN Technical Specification R0003-v06.00[32], Section 6.1, as shown in Eq. (8). This expression integrates all relevant system parameters, including PRB allocation, modulation scheme, coding rate, control overhead, and orthogonal frequency division multiplexing (OFDM) symbol timing, into a practical and measurable rate computation for both downlink and uplink scenarios.

$$R_{\mathrm{th}} = \sum_{j=1}^{J} \left( v_{\mathrm{Layer}}^{j}\, Q_m^{j}\, R_c\, \frac{12\, N_{\mathrm{PRB}}^{\mathrm{BW},j}}{T_s}\, (1 - OH^j)\, DLUL_{\mathrm{r}} \right), \tag{8}$$

where $J$ is the number of aggregated LTE component carriers, $R_c$ is the coding rate corresponding to channel quality, $v_{\mathrm{Layers}}$ is the number of MIMO layers, $Q_m$ is the modulation order, $N_{PRB}^{BW}$ is the number of PRBs allocated in bandwidth BW as per Table 2, $OH$ is overhead for control channels and signaling, $T_s$ is the average OFDM symbol duration and $DLUL_{\mathrm{r}}$ is a ratio of the symbols allocated for DL or UL data to total number of symbols.

In our implementation, the deployed system follows a single input single output (SISO) configuration in which the O-RAN base station is equipped with two physical antennas, one dedicated for downlink and the other for uplink, while both the haptic controller and the robotic arm are equipped with a single antenna each. As a result, the number of MIMO layers $v_{\mathrm{Layer}}$ is equal to 1, and only a single LTE component carrier is used ($J = 1$). Under these constraints, (8) simplifies to a more practical form suitable for our SISO-based teleoperation testbed, as shown in (9). This expression still incorporates modulation order $Q_m$, coding rate $R_c$, the number of allocated PRBs $N_{\mathrm{PRB}}^{\mathrm{BW}}$, OFDM symbol timing $T_s$, and overhead factors, but reflects a single-layer, single-carrier configuration. The simplified model provides a close estimation of achievable throughput under real testbed conditions and allows us to validate the impact of MCS configuration and PRB scheduling on the observed PHY performance during robotic

**Table 3 | The trace of srsRAN eNB DL metrics**

| rat | rnti | cqi | ri | mcs | brate | ok | nok | (%) |
|---|---|---|---|---|---|---|---|---|
| lte | 4f | 10 | 0 | 27 | 9.8M | 291 | 6 | 2% |
| lte | 4f | 10 | 0 | 27 | 11M | 323 | 10 | 3% |
| lte | 4f | 10 | 0 | 27 | 10.0M | 308 | 8 | 2% |
| lte | 4f | 10 | 0 | 26 | 11M | 338 | 17 | 4% |
| lte | 4f | 10 | 0 | 26 | 10M | 335 | 16 | 4% |
| lte | 4f | 10 | 0 | 26 | 11M | 346 | 16 | 4% |
| lte | 4f | 10 | 0 | 26 | 10M | 335 | 8 | 2% |
| lte | 4f | 10 | 0 | 27 | 10M | 323 | 10 | 3% |
| lte | 4f | 10 | 0 | 27 | 11M | 319 | 7 | 2% |
| lte | 4f | 10 | 0 | 27 | 11M | 325 | 7 | 2% |

**Table 4 | The trace of srsRAN eNB UL metrics**

| pusch | pucch | phr | mcs | brate | ok | nok | {(%) | bsr |
|---|---|---|---|---|---|---|---|---|
| 33.2 | 23.7 | 31 | 21 | 287k | 65 | 0 | 0% | 0.0 |
| 31.1 | 23.9 | 31 | 22 | 328k | 63 | 0 | 0% | 0.0 |
| 20.4 | 23.9 | 31 | 19 | 342k | 73 | 36 | 33% | 0.0 |
| 18.6 | 24.7 | 31 | 19 | 323k | 67 | 43 | 39% | 0.0 |
| 18.5 | 24.7 | 31 | 17 | 296k | 75 | 36 | 32% | 0.0 |
| 17.4 | 24.6 | 31 | 16 | 322k | 75 | 29 | 27% | 0.0 |
| 17.5 | 24.3 | 31 | 15 | 305k | 77 | 24 | 23% | 0.0 |
| 17.4 | 24.2 | 31 | 14 | 275k | 73 | 18 | 20% | 0.0 |
| 16.6 | 24.3 | 31 | 13 | 316k | 64 | 15 | 18% | 0.0 |
| 18.5 | 24.6 | 31 | 14 | 312k | 80 | 13 | 13% | 0.0 |

control sessions.

$$R_{\text{th}} = Q_m R_{\text{code}} \frac{12 N_{\text{PRB}}^{\text{BW}}}{T_s} (1 - OH) DLUL_{\text{ratio}}. \quad (9)$$

**Numerical validation of theoretical throughput in frequency division duplexing (FDD)-based O-RAN system.** Let us consider BW = 20 MHz, $N_{PRB}^{BW} = 100$, $OH$=0.15, MCS=28, the modulation 64 QAM, $Q_m = 6$, $R_c = 0.9258$ and $DLUL_{\text{ratio}} = 1$.

$$R_{\text{th}} = \left( 6 \times 0.9258 \times \frac{100 \times 12}{\frac{10^{-3}}{14}} (1 - 0.15) \times 1 \right) = 74.34 \, \text{Mbit s}^{-1}. \quad (10)$$

Equation (10), which calculates the PHY data rate, is adapted from the O-RAN TIFG E2E Test Specification[32]. While the original document includes an example using a DLUL_ratio of 0.543, likely corresponding to a time division duplexing system, we revise this to reflect our actual deployment using LTE Band 7 in FDD mode.

In FDD, uplink and downlink transmissions occur simultaneously on separate frequency bands. Therefore, all symbols allocated for the downlink are considered usable, and the DLUL_ratio term should be set to 1.

The inclusion of the analytical data transmission model in our paper was intended to complement the empirical results obtained from the physical testbed. Specifically, the model serves as a tool to understand the upper bounds of the achievable throughput based on radio parameters and PHY characteristics, and to explain how factors such as MCS, resource blocks, and control overhead affect link capacity.

While the setup is indeed based on a real end-to-end O-RAN-compliant system using srsRAN and USRP hardware, we used the model to relate the SINR to achievable rate through its effect on the modulation order

$Q_m$ and coding rate $R_c$. As mentioned in the O-RAN Alliance specifications[32], the SINR does not explicitly appear in the PHY throughput as in (8), but it indirectly determines the MCS thus influencing both $Q_m$ and $R_c$. Thus, our analysis is grounded in empirical measurements, and the data model serves as a complementary tool to understand how radio configuration parameters (e.g., PRB allocation, MCS) impact achievable throughput. Consequently, no synthetic channel model (e.g., Rayleigh fading) was used in the performance results. All results reflect actual physical wireless channel conditions within our test environment.

## Experimental results and discussion

In this study, we present the metrics obtained from the default trace in the command line interface tool for srsRAN. Tables 3 and 4 show the trace of the srsRAN eNB where metrics are provided for the DL and UL, respectively. In our current work, the iperf3 tool was primarily used to generate and analyze baseline LTE throughput under different radio configurations in a controlled and repeatable manner. This allowed us to validate the end-to-end system performance of the open-source srsRAN stack and USRP hardware under various MCS, while keeping the application-layer traffic consistent and quantifiable. We acknowledge that iperf3-generated traffic does not capture the full dynamics of real-time teleoperation data.

Beyond the iperf3 traffic, our experimental framework was also used with real haptic command sequences and feedback signals between the operator and the robot. The haptic stream includes time-series data such as position vectors and force feedback sampled at 100–1000 Hz, encapsulated in LTE packets and exchanged over the deployed Open RAN network. These sequences were streamed using a real robotic teleoperation interface, similar in spirit to prior work such as in[2].

Additionally, we utilize the Iperf3 command to assess network performance metrics such as bandwidth (bitrate) and data transfer. The data transfer, measured in Megabytes (MB), indicates the amount of data successfully transmitted within each interval (e.g., 0−1 s, 1−2 s, etc.). The bitrate, measured in Megabits per second (Mbps), represents the network's transmission speed. We generated traffic for 3 min from the O-RAN server to the srsUE haptic controller and srsUE robotic arm (downlink direction) to evaluate performance over time for both haptic control and robotic arm, with a consistent observation window of 1 s. The transfer rates were predominantly between 1.12 MB/s and 1.25 MB/s, corresponding to bitrates ranging from 9.44 Mbps to 10.5 Mbps. The bitrate and data transfer showed slight fluctuations within a ±1 Mbps range, alternating between 9.44 Mbps and 10.5 Mbps, suggesting a periodic or cyclic pattern. Occasional drops (e.g., 9.44 Mbps) were observed every few seconds, likely due to interference from other users and sources, as noted in Eq. (5). Despite these fluctuations, the communication link remained stable, with no significant drops, losses, or outliers over the 180 s. We captured a snapshot of the performance for 5 s to illustrate the bitrate and data transfer for both the haptic controller and robotic arm as per Fig. 4. The USRP and the dongle consistently maintain stable bitrate around 10 Mbps. Both devices show comparable average overall performance.

Figure 5 illustrates the relationship between MCS values and the corresponding bitrate for both DL and UL directions for the robotic arm user. While MCS DL reaches a peak value of 27 during high-SINR conditions, the log traces confirm that it periodically drops to values between 20 and 21 under interference or signal degradation.

The DL bitrate remains near 10 Mbps when MCS DL = 27, but significantly declines to less than 2 Mbps when MCS values decrease. This validates the expected dependency between MCS and achievable throughput. We have revised the plotted time window of Fig. 5 to better reflect this variation.

To avoid oversimplification, we have also separated DL and UL discussions. The UL MCS remains within a narrower range (15–21), resulting in consistently lower throughput due to uplink path loss and lower scheduling priority. This asymmetry highlights the importance of analyzing DL and UL behaviors independently in real-time teleoperation scenarios.

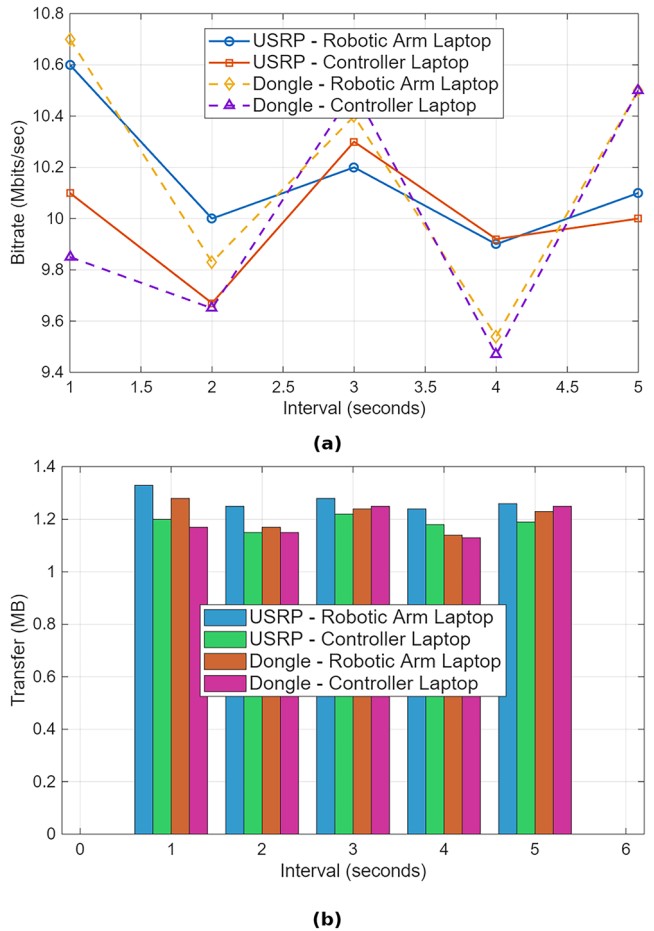

**Fig. 4 | Comparison of downlink bitrate and data transfer for USRP and mobile dongle. a** Bitrate (Mbits/sec) performance evaluated at the robotic arm and haptic controller laptops over five intervals. **b** Corresponding data transfer (MB) across the same intervals.

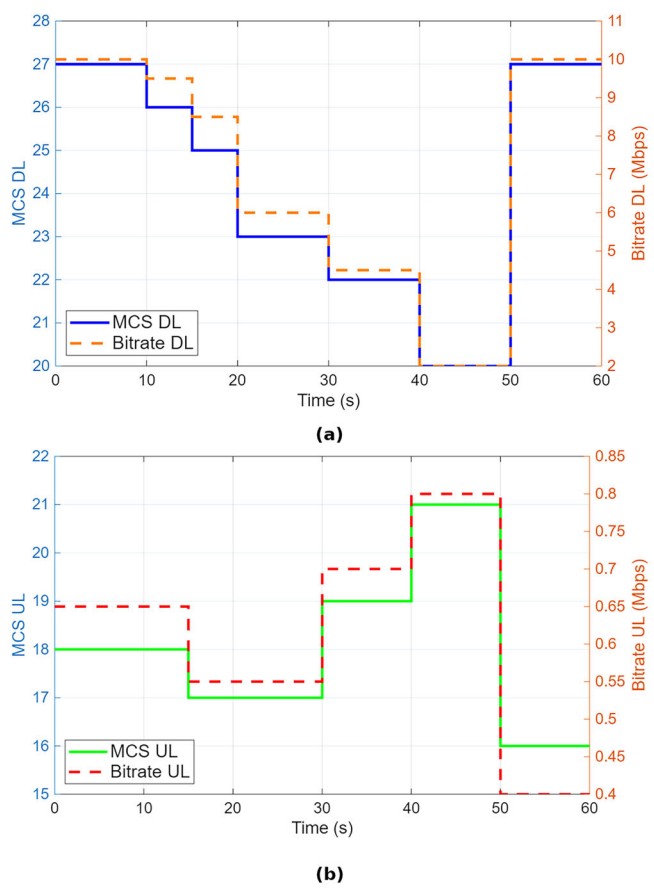

**Fig. 5 | Time evolution of modulation and coding scheme (MCS) and bitrate for the robotic arm user. a** Downlink (DL) transmission: bitrate varies from 10 to 2 megabits per second (Mbps) as the MCS level decreases from 27 to 20. **b** Uplink (UL) transmission: bitrate ranges between 0.4 and 0.8 Mbps as the MCS level fluctuates between 13 and 21. The figure illustrates the adaptive behavior of DL and UL communication over a 60-s evaluation window.

Consequently, the BLER increases as the MCS values increase as per Fig. 6. The custom xApp is developed to identify network interference and dynamically modify the MCS to counteract its effects. The custom xApp leverages a Kubernetes-based deployment using Docker containers and integrates a rules-based heuristic detector for real-time interference classification. We used the implementation[25] which adopts lightweight classifiers (such as decision trees or logistic regression) for distinguishing interference types based on features extracted from RIC telemetry data (e.g., reference signal received power, SINR, and throughput). We would like to emphasize that this custom xApp was successfully integrated and tested in our fully deployed end-to-end 4G LTE system. This system is built using open-source srsRAN and USRP hardware, and supports a teleoperation use case, where low-latency and reliable communication are critical. In this practical setup, we found that the use of advanced algorithms, such as reinforcement learning or context-aware optimization, was not necessary, as long as system performance metrics such as BLER and latency remained within acceptable thresholds. The xApp sends a command to start using adaptive MCS, the trace will show the MCS values changing to around 13–21, according to the signal quality.

To gain deeper insights into the O-RAN network, we deployed the multi-user KPIMON xApp to monitor RIC indication messages exchanged between the RIC hosting the xApp and the network nodes (e.g., eNBs, gNBs, or specific RAN elements like DU and CU). The KPIMON xApp is designed to gather network performance metrics using the E2SM-KPM Service Model, periodically collecting the KPM from the RAN. In our setup, the collected KPMs include the RAN ID, the number of active UE

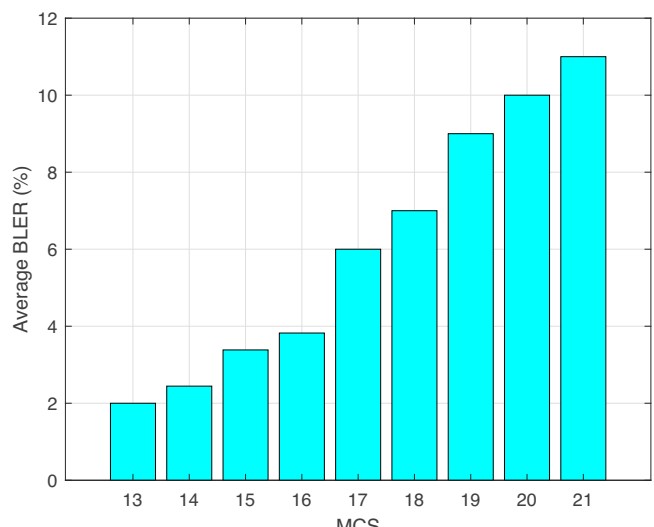

**Fig. 6 | Block Error Rate (BLER) performance across different Modulation and Coding Scheme (MCS) indices.** The figure shows the average BLER measured in the uplink (UL) direction for MCS indices ranging from 13 to 21. Higher MCS values lead to increased BLER, highlighting the trade-off between throughput and link reliability in the O-RAN-based system.

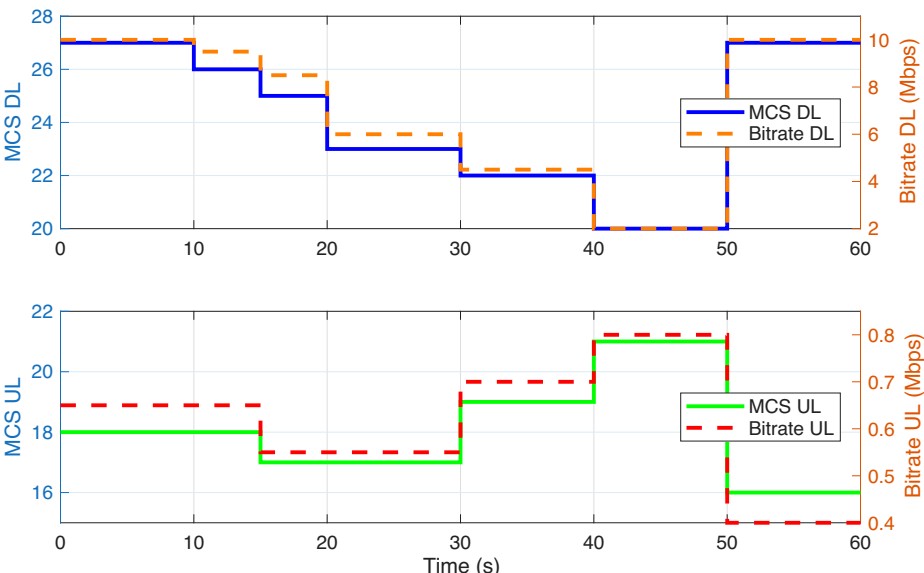

**Fig. 7 | Uplink Packet Data Convergence Protocol (PDCP) throughput over time.** The figure illustrates the PDCP bytes received at the Evolved Node B (eNB) from the Software Radio Systems User Equipment (srsUE) haptic controller and the srsUE robotic arm. Throughput is measured over consecutive 1-s intervals, showing stable uplink performance of ~10 Mbps per UE in the deployed O-RAN system.

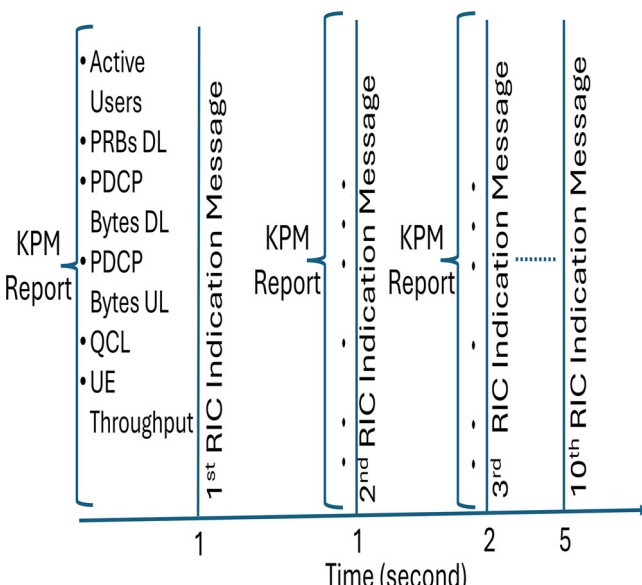

**Fig. 8 | Cycle of RAN Intelligent Controller (RIC) indication messages over time.** The figure shows periodic Key Performance Measurement (KPM) reports including active users, Physical Resource Blocks (PRBs), Packet Data Convergence Protocol (PDCP) bytes, Quality of Service Class Identifier (QCI), and throughput metrics.

connections, QoS class identifiers, and the amount of downlink and uplink data transmitted by the gNB in bytes. This Service Model supports only the report service type, meaning it relies solely on indication messages from the RAN.

Figure 7 illustrates the data received in bytes by the eNB from the srsUE haptic controller and the srsUE robotic arm connected to it for a specific QCI. In this scenario, two UEs are considered, transmitting uplink data at rates of 10 Mbps for each UE. Each radio component, including the eNB and the two UEs, utilizes an X310 USRP paired with its respective compute node. The KPIMON SM monitors and collects the total PDCP bytes for srsUE haptic and srsUE robotic arm while the srsUE haptic/robotic arm remains connected to the eNB. The periodic event trigger timer is configured to 1 s.

Figure 8 shows the messages are received in quick succession around the same interval of 1 s, reflecting a rapid cycle of data collection and

processing. The repeated behavior shows up-to-date information on network performance and resource usage. The fact that two RIC Indication Messages are happening within the same second suggests low-latency communication. The time difference between two consecutive messages could be milliseconds.

Figure 9 shows that there are occasional spikes in latency (e.g., at 30 s), but the overall packet loss remains low throughout, suggesting that the system is relatively stable even during moments of higher latency. This is a good sign for real-time or near-real-time network monitoring and optimization, especially in critical use cases like our use case in this research which is robotic arm operation for denture inspection.

We clarify that "Latency (ms)" is application layer round-trip time measured at 1s granularity, while "Packet Loss (%)" is computed over the same 1s bins. Because the two series are sampled rather than per packet and are affected by different buffering/retransmission processes, contemporaneous peaks need not coincide. In LTE, HARQ (PHY/MAC) and RLC acknowledged mode (AM) retransmissions can queue and delay packets while still delivering them successfully, producing latency spikes with little or no loss. HARQ rounds are specified at MAC (3GPP TS36.321), and RLC AM recovery, including t Reordering/t Poll Retransmit timers, is specified in RLC (3GPP TS36.322). When these mechanisms are active (e.g., transient radio impairments), end-to-end delay can increase noticeably without producing drops at the application layer. Conversely, when discard timers/policies at RLC AM/PDCP expire (e.g., if recovery exceeds a configured bound), packets may be dropped promptly, yielding loss spikes with only a small contemporaneous latency increase. PDCP/RLC timer behavior (e.g., PDCP reordering/discard, RLC AM STATUS timing) is standardized (TS36.323/TS36.322); operational discussions widely note that a tighter PDCP discard window can surface as loss even when mean latency stays moderate.

**Theoretical vs. experimental throughput**

While Eqs. (8)–(10) provide a theoretical model for physical-layer throughput based on modulation order $Q_m$, coding rate $R_c$, PRB allocation, and OFDM symbol timing, they represent idealized upper bounds. These expressions assume full PRB utilization, perfect link conditions, negligible overhead, and no scheduler inefficiencies.

In practice, our experimental results, collected via iperf3 at the application layer, demonstrate significantly lower throughput values due to various system-level and hardware constraints.

For example, using 100 PRBs, 64-QAM ($Q_m = 6$), and $R_c \approx 0.9258$, the theoretical downlink throughput reaches ~74.34 Mbps. However, as shown

**Fig. 9 | Real-time measurements of network latency and packet loss over a 40-s test window.** Latency (blue line) is measured in milliseconds (ms), and packet loss (red line) is shown as a percentage %, highlighting the transient behaviors during initial link setup and stabilization.

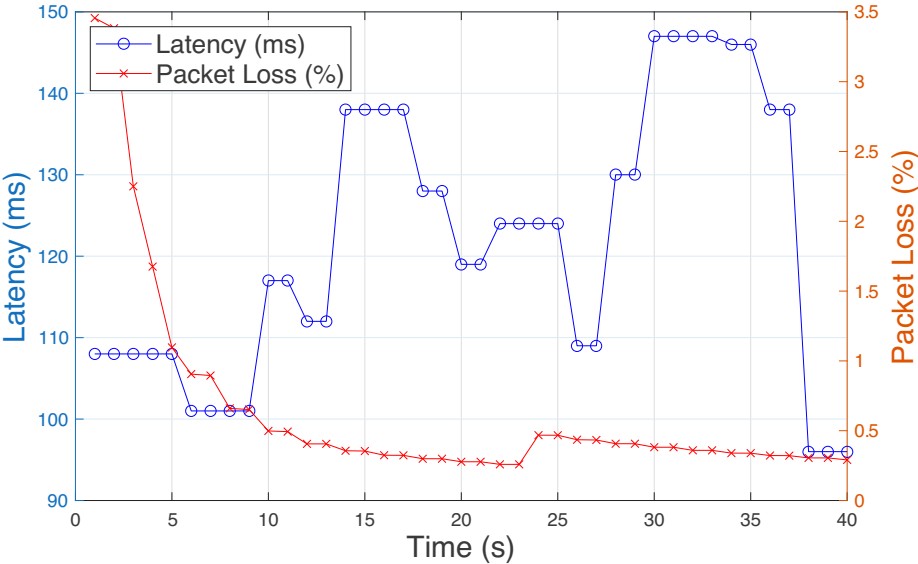

## Table 5 | Comparison of theoretical and measured throughput

| Metric | Theoretical | Measured | Notes |
|---|---|---|---|
| MCS index | 28 (64-QAM) | 18-27 (DL) | Adaptively lowered based on SINR variations |
| PRBs used | 100 | Partial | Periodic gaps in scheduling |
| DL throughput | 74.34 Mbps | 9.4–10.5 Mbps | Measured with iperf3. Transmission initiated with: sudo iperf3 -c 162.16.0.1 -p 5006 -i 1 -t 180 -R -b 10M, targeting 10 Mbps for benchmarking against theoretical maximum |
| UL MCS index | N/A | 13–21 | Limited by uplink link quality |
| UL throughput | N/A | 0.3–0.4 Mbps | Affected by conservative MCS and traffic profile |
| BLER | ~ 0% (ideal) | 6–12% | See Fig. 6; increases with higher MCS |
| PDCP byte trend | N/A | Periodic bursts | Matches KPIMON event triggers every 1s |

in Fig. 5, the observed downlink throughput for both the haptic controller and robotic arm ranges between 9.4 and 10.5 Mbps. Uplink throughput is even lower, limited to 0.3–0.4 Mbps due to lower MCS values and protocol overhead.

This discrepancy is expected and primarily results from:
- MAC-layer inefficiencies and partial PRB utilization, please refer to Table 2.
- Control and RRC signaling overheads.
- Buffering delays and software radio latency in the srsRAN stack.
- Retransmissions and imperfect SINR conditions during indoor experiments.
- Periodic PDCP byte transmission patterns, confirmed in Fig. 8.To summarize the comparison, we provide Table 5.

This section helps clarify the practical gap between spec-based models and observed results, while reinforcing the relevance of empirical validation.

## Conclusion

This study successfully demonstrates the real-time integration of a robotic arm within O-RAN network tailored for dental inspection, showcasing the viability of low-latency, cost-effective solutions in healthcare robotics. By developing an open-source O-RAN testbed with SDRs, we achieved seamless interoperability between a haptic controller, robotic arm, and the near-real-time RAN RIC, achieving critical sub-one-second latency for remote dental procedures. The replacement of traditional USRP SDRs with customized mobile dongles (Huawei EE32) reduced power consumption by 90% (4.5 W vs. 45 W) and hardware costs by orders of magnitude, enabling scalable and dynamic deployments. Stable communication performance, maintaining ~10 Mbps bitrate with minimal packet loss, validated the

robustness of the O-RAN architecture under adaptive MCS. The integration of xApps (e.g., KPIMON for monitoring and a custom xApp for MCS adaptation) further highlighted the flexibility of O-RAN's open interfaces for real-time control and optimization.

This work not only advances the practical implementation of O-RAN in healthcare robotics but also establishes a foundational framework for future research into AI-driven network optimization, multi-robot coordination, and expanded use cases requiring ultra-reliable, low-latency communication. In future iterations, integrating native 5G stacks with matured E2/O1 capabilities could further enhance scalability and mobility features. Moreover, the modular testbed developed herein offers a reproducible platform for experimentation across industrial automation, telesurgery, and emergency response scenarios.

## Data availability

Correspondence and requests for data should be addressed to Qammer H. Abbasi.

## Code availability

The software used to implement our Open RAN-based robotic teleoperation testbed is publicly available through the Open AI Cellular (OAIC) GitHub repositories. Specifically: • E2 Agent-enabled srsRAN (4G srsLTE stack): https://github.com/openaicellular/srslte-e2, which enables E2 integration with the Near-RT RIC. • OAIC core platform and deployment scripts: https://github.com/openaicellular/oaic, which includes installation guides, RIC deployment, and xApp onboarding scripts. Detailed guides on developing and deploying custom xApps (e.g., ML prototypes) are available via OAIC's documentation: https://openaicellular.github.io/oaic/quickstart.

htmland https://openaicellular.github.io/oaic/xapp_python.html. All dataset collection scripts (e.g., iperf3 sequencing, telemetry log extraction), performance visualization tools, and testbed orchestration code are available upon reasonable request to the corresponding author. Note that our robotic teleoperation control scripts depend on a proprietary haptic interface and therefore cannot be openly shared, but their functional operation is described in Methods, and their behavior can be reproduced with commercially available substitutes.

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

## Acknowledgements

This work is supported under CHEDDAR EP/X040518/1 and EPSRC UK-India Future Networks Initiative (Grant ref- EP/W016524/1) led by the University of East Anglia in the UK.

## Author contributions

S. Hassouna: Conception and design of study, Simulation/Experiment, Acquisition of data, Writing - original draft. Jaspreet Kaur, Burak Kizilkaya, Jalil ur Rehman Kazim, Shuja Ansari: Conception and design of study, Acquisition of data. Arzad Alam Kherani, Brijesh Lall: reviewing and proofreading. M. A. Imran: Analysis and/or interpretation of data, review and editing original draft. Q. H. Abbasi: Analysis and/or interpretation of data, review and editing original draft.

## Competing interests

The authors declare no competing interests.
