## [Transparent Peer Review file · Communications Engineering]

Development of Open RAN for Real-time Teleoperation

Corresponding Author: Professor Qammer Abbasi

Version 0:

Reviewer comments:

Reviewer #1

(Remarks to the Author)

This paper presents an O-RAN testbed design with robotic end devices, i.e., a haptic controller and a robotic arm. The use case is teleoperation for dental inspections. The system is designed using open-source libraries and software. Experiments show transmission stability in both the Uplink and the Downlink. In terms of latency, the system could operate near-real-time, showing the potential to be used in teleoperations. Some questions and suggestions about this work are:

1. Would you consider making your implementation open-source? It would be a valid contribution to the community.
2. Fig. 4 is not informative and hard to read. Please consider changing it to a table instead.
3. On page 6, it mentioned that "This implementation requires deploying a Kubernetes-based environment with Docker containers, incorporating machine learning models for interference detection". Could you elaborate on the machine learning models? Was that some automated MSC adaptation methods? Would you consider improving the MSC adaptation algorithms?

Reviewer #2

(Remarks to the Author)

This is a testbed design paper where the authors are implementing the O-RAN testbed for teleoperation through two UEs. Overall the idea is very relevant and implementation. There have been a few papers which has proposed similar ideas which he authors have also acknowledged.

I think this paper has a potential to make impact and influence the field. However, I am a bit concerned about the aspect of it. There are a few concerns which are not clear from this version:

1. What is the overall framework? A representative figure of each block would have been helpful. What type of data is being transmitted during the experiment?
2. The usecase specific analysis, rather than randomly transmitting traffic through iperf, is it possible to generate a few haptic sequence for experiment setting, and evaluate them. A more organized experiment set would be very helpful for potentially establishing such idea.
3. Not clear how the data transmission model is relevant - why we need intermediate Rayleigh channels, how about experiment on real channel in various settings? After all this is a real setup.

Reviewer #3

(Remarks to the Author)

Review Comments

1. Summary of the Paper

This paper presents an end-to-end 4G LTE open-source system based on srsRAN, integrated with the O-RAN Near-RT RIC and two xApps: KPIMon and a custom xApp displaying MCS and BLER metrics. The system is evaluated in a teleoperation scenario involving a robotic arm and a haptic controller.

However, the work does not demonstrate any novel utilization of O-RAN capabilities or significant use of O-RAN open

interfaces tailored for teleoperation applications. Additionally, deploying low-latency robotic control over 4G LTE is not state-of-the-art, particularly considering the availability of 5G technology. With the same USRP X310 hardware, the authors could have deployed the srsRAN 5G stack and tested with a 5G dongle or srsUE stack to better align with modern teleoperation requirements.

Furthermore, there is no compelling justification for deploying the custom xApp solely to monitor MCS and BLER, especially in the absence of any concrete control actions taken by the xApp to improve teleoperation performance (e.g., latency reduction or reliability enhancement).

2. Strengths of the Paper

- The authors successfully deployed a complete end-to-end 4G LTE system using open-source srsRAN and USRP hardware, applying it to a teleoperation use case involving a robotic arm and a haptic controller.

3. Areas for Improvement

- Section III – O-Cloud Terminology: The statement "The O-Cloud comprises two primary units..." is misleading. If the O-CU and O-DU are deployed on bare-metal servers, this does not constitute an O-Cloud. The term O-Cloud is appropriate only when referring to a virtualized or cloud-native deployment environment.
- Formulae (1)-(7): The relevance of formulae (1) through (7) is unclear. Ultimately, the paper uses formula (8), which is derived from 3GPP or O-RAN specifications. However, the citation "[24] O-RAN website" is vague and unhelpful. The authors should cite the exact technical document (e.g., O-RAN TIFG E2E Test R0003-v06.00, Section 6.1) to support the formula.
- Theoretical vs. Experimental Throughput: There is a significant discrepancy between the theoretical throughput (as derived from formulas (8)–(10)) and the actual throughput observed in Section IV. The authors should clarify the purpose of presenting the theoretical throughput and provide explanations for the observed differences.
- DLUL Ratio in Formula (10): The use of $DLUL_ratio = 0.543$ is incorrect for LTE Band 7 in FDD mode, where uplink and downlink operate on different center frequencies. This ratio is typically used in TDD mode. For FDD, the $DLUL_ratio$ should be 1.
- Figure 6 – MCS and Bitrate Discrepancy: In Figure 6 (top), only MCS DL = 27 is shown, which appears to be an error. Even if MCS DL varies from 20 to 27, the corresponding DL bitrate fluctuates around 10 Mbps, which contradicts the observations described in Section IV. Also, combining UL and DL observations into a single sentence is misleading and oversimplifies the system behavior.
- Custom xApp Functionality: The role and operation of the custom xApp require more detailed explanation. The statement that it can "identify network interference and dynamically modify the MCS to counteract its effects" is vague and raises several important questions:
 - o Did the authors modify the MCS selection mechanism in the eNB? According to 3GPP standards, the eNB determines MCS based on CQI feedback from the UE, considering CSI-RS, SINR, BLER, and HARQ performance.
 - o If the xApp overrides this process, how does that affect UE behavior and overall system compatibility?
- Lack of Closed-Loop Control: Overall, the purpose of the custom xApp remains unclear. Without a closed-loop system—where KPIMon monitors performance and the xApp applies control commands based on optimization within the Near-RT RIC—the use of a custom xApp lacks justification.

Version 2:

Reviewer comments:

Reviewer #1

(Remarks to the Author)

My concerns are addressed. Recommend acceptance.

Reviewer #2

(Remarks to the Author)

The authors have addressed all my comments and improved the revised article significantly. I do not have any further major suggestions.

Just do a final typo and grammatical error checking -

Typo: 'block error rate (Bler)' -> 'BLER' at page 7.

Full form is not mentioned at the first occurrence of BLER - at abstract also in the introduction. First time it is expanded is in section II, page 3.

Please perform a thorough typo checking.

Reviewer #3

(Remarks to the Author)

Comments:

- The authors have addressed most of the critical points raised in my previous review. I appreciate their efforts in improving

the manuscript.

Questions and Clarifications:

1. srsRAN 5G Stack Limitations

The authors claim that the srsRAN 5G stack exhibits limitations in terms of stability and full-stack integration. Could the authors elaborate on the specific aspects or components where these limitations are observed?

2. RIC Integration Support

According to the srsRAN website and GitHub repository, RIC integration is natively supported in the 5G stack. Could the authors clarify what additional integration or modification was required in their implementation?

3. CU/DU Split in srsRAN 4G Stack

The concept of CU/DU disaggregation is commonly associated with 5G. In my understanding, srsRAN's 4G stack (i.e., srsLTE) compiles the eNB as a single monolithic instance, without explicit CU/DU split. Could the authors provide a credible reference or documentation to support the claim that CU/DU disaggregation is available in the 4G stack? If this is a novel contribution or workaround, more details would be helpful.

4. srsUE Limitations in 5G SA Mode

The authors mention that srsUE has constraints in 5G SA mode, such as limited bandwidth support and lack of handover functionality.

- Why does srsUE need to distinguish between SA and NSA mode, given its role as a user equipment?

- Could the authors clarify how bandwidth limitations manifest in 5G compared to 4G for srsUE?

- Please provide references or documentation to support the claim about lack of handover support in srsUE.

5. Stability Issues in srsRAN 5G Stack

The authors mention observing stability issues such as UE disconnections after a few minutes. Could they provide more concrete evidence or references for this observation?

Based on my own experience with the srsRAN 5G stack (as of 2024), it was relatively stable, and recent versions have shown further improvements. This claim seems surprising and warrants further clarification.

6. Choice of xApp over rApp

Why did the authors choose to implement the monitoring component as an xApp? Are there technical or architectural reasons that make xApp preferable over an rApp in this context? I do not see value of monitoring sub-second performance metrics with xApp.

7. ML Logic in Custom xApp

Regarding the ML-based xApp:

- What type of ML task is being used—classification, regression, or something else?

- How was the model trained, and on what data?

- What is the specific objective of using ML here?

If the xApp only identifies degradation patterns in metrics such as MCS or BLER using simple rules, it is unclear why an ML model is needed. Further justification is required.

8. Throughput Discrepancy Explanation

The explanation for the discrepancy between theoretical and measured throughput is vague and lacks supporting evidence. The reasons provided are too generic and implicitly suggest that srsRAN 4G's implementation is subpar, which needs to be substantiated with data or references.

10. Figure 9 – Latency vs. Packet Loss

The correlation between latency and packet loss in Figure 9 is unclear. For example, in the time range 0–5s, packet loss is high (up to 3.5%) while latency remains low and stable (~110ms). Conversely, when packet loss drops below 1%, latency becomes more volatile. The authors are encouraged to re-examine this relationship or provide an explanation for this discrepancy.

Version 3:

Reviewer comments:

Reviewer #3

(Remarks to the Author)

The authors have addressed most of the critical points raised in my previous review. I appreciate their efforts in improving the manuscript.

Author's Response Letter:

Manuscript ID: manuscript COMMS-ENG-25-0245

Paper Title: Development of Open RAN for Real-time Teleoperation

Authors: Saber Hassouna, Jaspreet Kaur , Burak Kizilkaya, Jalil Kazim, Shuja Ansari, Arzad Alam Kherani, Brijesh Lall, Qammer H. Abbasi, and Muhammad Ali Imran

The authors would like to thank the Editor and the Reviewers for their time and valuable comments. We have revised our manuscript and have addressed the specific comments of the reviewers and the editor below in this response letter. We have highlighted the changes in the blue font color in the revised version of the manuscript and the reasonings for these changes are given in the responses below.

We hope that the concerns of the Editor and the Reviewers have been appropriately addressed in the revised version and the paper is accepted for publication in this form.

EDITOR'S COMMENTS:

General Comments: *Your manuscript entitled "Development of Open RAN for Real-time Teleoperation" has now been seen by 3 referees, whose comments are appended below. In the light of their advice I regret to inform you that we cannot publish your manuscript in Communications Engineering.*

While Reviewers 1 and 2 noted the potential relevance of your O-RAN design for teleoperation and offered suggestions (e.g., open-sourcing the implementation, refining figures, and enhancing experimental design), the most important feedback comes from Reviewer 3, whose detailed technical evaluation highlights substantial shortcomings that preclude the paper from advancing the field at this stage. We acknowledge the effort invested in this work, unfortunately, the lack of a clear innovative contribution to O-RAN teleoperation and the unresolved technical flaws are sufficiently important to preclude publication of this study in Communications Engineering.

I am sorry that we cannot be more positive on this occasion and thank you for the opportunity to consider your work.

Response: We would like to thank the Editor for the careful review of our manuscript. We have made extensive changes and thoroughly revised our manuscript, according to the useful feedback received by the Editor and the Reviewers. The changes are duly highlighted in blue font color in the revised version of the manuscript.

Finally, we would like to thank the Editor for constructive comments that helped us to considerably improve the technical and presentation quality of our manuscript.

REVIEWER 1'S COMMENTS:

General Comments: *This paper presents an O-RAN testbed design with robotic end devices, i.e., a haptic controller and a robotic arm. The use case is teleoperation for dental inspections. The system is designed using open-source libraries and software. Experiments show transmission stability in both the Uplink and the Downlink. In terms of latency, the system could operate near-real-time, showing the potential to be used in teleoperations. Some questions and suggestions about this work are:*

Response: Authors are grateful to the reviewer for the positive feedback and critical review of the manuscript.

Comments 1: *Would you consider making your implementation open-source? It would be a valid contribution to the community.*

Response 1: Authors are thankful to the reviewer for the valuable comment. In response to this valuable comment, we would like to clarify and confirm that when our work is accepted for publication, we will make it open source to the public.

Comment 2: *Fig. 4 is not informative and hard to read. Please consider changing it to a table instead.*

Response 2: Authors are thankful to the reviewer for the valuable comment. In response to this valuable comment, please refer to our below highlighted response in blue font colour in the revised manuscript at section IV (Experimental results) – page 7 and we highlight it below as well:

In this study, we present the metrics obtained from the default trace in the command line interface (CLI) tool for srsRAN. Tables III and IV show the trace of the srsRAN eNB where metrics are provided for the DL and UL respectively.

TABLE III
THE TRACE OF SRSRAN GNB DL METRICS

rat	rnti	cqi	ri	mcs	brate	ok	nok	(%)
lte	4f	10	0	27	9.8M	291	6	2%
lte	4f	10	0	27	11M	323	10	3%
lte	4f	10	0	27	10.0M	308	8	2%
lte	4f	10	0	26	11M	338	17	4%
lte	4f	10	0	26	10M	335	16	4%
lte	4f	10	0	26	11M	346	16	4%
lte	4f	10	0	26	10M	335	8	2%
lte	4f	10	0	27	10M	323	10	3%
lte	4f	10	0	27	11M	319	7	2%
lte	4f	10	0	27	11M	325	7	2%

TABLE IV
THE TRACE OF SRSRAN GNB UL METRICS

pusch	pucch	phr	mcs	brate	ok	nok	(%)	bsr
33.2	23.7	31	21	287k	65	0	0%	0.0
31.1	23.9	31	22	328k	63	0	0%	0.0
20.4	23.9	31	19	342k	73	36	33%	0.0
18.6	24.7	31	19	323k	67	43	39%	0.0
18.5	24.7	31	17	296k	75	36	32%	0.0
17.4	24.6	31	16	322k	75	29	27%	0.0
17.5	24.3	31	15	305k	77	24	23%	0.0
17.4	24.2	31	14	275k	73	18	20%	0.0
16.6	24.3	31	13	316k	64	15	18%	0.0
18.5	24.6	31	14	312k	80	13	13%	0.0

Comment 3: *On page 6, it mentioned that "This implementation requires deploying a Kubernetes-based environment with Docker containers, incorporating machine learning models for interference detection". Could you elaborate on the machine learning models? Was that some automated MSC adaptation methods? Would you consider improving the MSC adaptation algorithms?*

Response 3: Authors are thankful to the reviewer for the valuable comment. In response to this valuable comment, please refer to our below highlighted response in blue font colour in the revised manuscript at section IV (Experimental results) – page 9 and we highlight it below as well:

The custom xApp leverages a Kubernetes-based deployment using Docker containers and integrates machine learning (ML) models for real-time interference classification. We used the implementation [25] which adopts lightweight ML classifiers (such as decision trees or logistic regression) for distinguishing interference types based on features extracted from RIC telemetry data (e.g., RSRP, SINR, and throughput).

We would like to emphasize that this custom xApp, with its simple ML model, was successfully integrated and tested in our fully deployed end-to-end 4G LTE system. This system is built using open-source srsRAN and USRP hardware, and supports a teleoperation use case — where low-latency and reliable communication are critical. In this practical setup, we found that the use of advanced or complex ML algorithms, such as reinforcement learning or context-aware optimization, was not necessary, as long as system performance metrics such as block error rate (BLER) and latency remained within acceptable thresholds. The xApp sends a command to start using adaptive MCS, the trace will show the MCS values changing to around 13-21, according to the signal quality.

Finally, we would like to thank the Reviewer for constructive comments that helped us to considerably improve the technical and presentation quality of our manuscript.

REVIEWER 2'S COMMENTS:

General Comments: *This is a testbed design paper where the authors are implementing the O-RAN testbed for teleoperation through two UEs. Overall the idea is very relevant and implementation. There have been a few papers which has proposed similar ideas which he authors have also acknowledged.*

I think this paper has a potential to make impact and influence the field. However, I am a bit concerned about the aspect of it. There are a few concerns which are not clear from this version:

Response: Authors are grateful to the reviewer for the positive feedback and critical review of the manuscript.

Comment 1: What is the overall framework? A representative figure of each block would have been helpful. What type of data is being transmitted during the experiment?

Response 1: We thank the reviewer for this insightful comment. please refer to our below highlighted response in blue font colour in the revised manuscript at section II (O-RAN Network Design and Implementation) – page 3 and we highlight it below as well:

Addressing the request regarding the overall framework and the representative figures:

The proposed system architecture is shown in Fig. 1 of the manuscript, which provides a high-level overview of the Open RAN-based framework integrating robotic control and haptic feedback via a fully deployed 4G LTE testbed. This includes the near-real-time RIC, the custom xApp, and the end-to-end connectivity between the operator-side haptic interface and the teleoperator robotic arm. Fig. 2 further clarifies the experimental testbed setup, including the servers, USRPs, mobile dongles, and UE hardware and Fig. 3 illustrates the physical deployment of the base station setup and connected devices.

Addressing the type of transmitted data:

The system transmits real-time control commands, such as position and velocity vectors from the haptic interface to the robot, and force feedback signals in the reverse direction. These are encapsulated in standard IP packets over LTE and routed through the srsRAN-based EPC. In addition, the iperf3 was used to generate UDP traffic in some test phases to characterize the LTE throughput under controlled load conditions, as described in the OAI Cellular documentation [20].

Comment 2: *The use case specific analysis, rather than randomly transmitting traffic through iperf, is it possible to generate a few haptic sequence for experiment setting, and evaluate them. A more organized experiment set would be very helpful for potentially establishing such idea.*

Response 2:

We thank the reviewer for this insightful comment. please refer to our below highlighted response in blue font colour in the revised manuscript at section IV (experimental results) – page 7/8 and we highlight it below as well:

In our current work, the iperf3 tool was primarily used to generate and analyze baseline LTE throughput under different radio configurations in a controlled and repeatable manner. This allowed us to validate the end-to-end system performance of the open-source srsRAN stack and USRP hardware under various MCS, while keeping the application-layer traffic consistent and quantifiable. We acknowledge that iperf3-generated traffic does not capture the full dynamics of real-time teleoperation data.

Beyond the iperf3 traffic, our experimental framework was also used with real haptic command sequences and feedback signals between the operator and the robot. The haptic stream includes time-series data such as position vectors and force feedback sampled at 100–1000 Hz, encapsulated in LTE packets and exchanged over the deployed Open RAN network. These sequences were streamed using a real robotic teleoperation interface, similar in spirit to prior work such as in [2].

Comment 3: *Not clear how the data transmission model is relevant - why we need intermediate Rayleigh channels, how about experiment on real channel in various settings? After all this is a real setup.*

Response 3:

We thank the reviewer for this insightful comment. please refer to our below highlighted response in blue font colour in the revised manuscript at section III (Data Transmission Model) – page 7 and we highlight it below as well:

The inclusion of the analytical data transmission model in our paper was intended to complement the empirical results obtained from the physical testbed. Specifically, the model serves as a tool to understand the upper bounds of the achievable throughput based on radio parameters and physical layer characteristics, and to explain how factors such as MCS, resource blocks, and control overhead affect link capacity.

While the setup is indeed based on a real end-to-end O-RAN-compliant system using srsRAN and USRP hardware, we used the model to relate channel quality (SINR) to achievable rate through its effect on the modulation order Q_m and coding rate R_c . As mentioned in the O-RAN Alliance specifications [28], the SINR does not explicitly appear in the physical layer throughput equation, but it indirectly determines the modulation and coding scheme (MCS) thus influencing both Q_m and R_c . Thus, our analysis is grounded in empirical measurements,

and the data model serves as a complementary tool to understand how radio configuration parameters (e.g., PRB allocation, MCS) impact achievable throughput. Consequently, no synthetic channel model (e.g., Rayleigh fading) was used in the performance results. All results reflect actual physical wireless channel conditions within our test environment.

Finally, we would like to thank the Reviewer for constructive comments that helped us to considerably improve the technical and presentation quality of our manuscript.

REVIEWER 3'S COMMENTS:

Summary of the Paper: *This paper presents an end-to-end 4G LTE open-source system based on srsRAN, integrated with the O-RAN Near-RT RIC and two xApps: KPIMon and a custom xApp displaying MCS and BLER metrics. The system is evaluated in a teleoperation scenario involving a robotic arm and a haptic controller.*

However, the work does not demonstrate any novel utilization of O-RAN capabilities or significant use of O-RAN open interfaces tailored for teleoperation applications.

Additionally, deploying low-latency robotic control over 4G LTE is not state-of-the-art, particularly considering the availability of 5G technology. With the same USRP X310 hardware, the authors could have deployed the srsRAN 5G stack and tested with a 5G dongle or srsUE stack to better align with modern teleoperation requirements.

Furthermore, there is no compelling justification for deploying the custom xApp solely to monitor MCS and BLER, especially in the absence of any concrete control actions taken by the xApp to improve teleoperation performance (e.g., latency reduction or reliability enhancement).

Response: We appreciate the reviewer's thoughtful feedback and the opportunity to clarify the contributions of our work. Please refer to our below highlighted response in blue font colour in the revised manuscript at section I (Introduction) – page 3 and we highlight it below as well:

Our primary objective was to demonstrate the feasibility of deploying an open, standards-aligned, end-to-end O-RAN system using open-source platforms (srsRAN and OpenAirInterface components) for real-time robotic teleoperation. While it is true that the use of 5G would provide lower latency and higher reliability, our motivation for selecting a 4G LTE stack was grounded in practical accessibility and current deployment maturity.

The open-source srsRAN 5G stack, while evolving, currently exhibits limitations in stability and full-stack integration, particularly at the UE and RIC interface layers, required for consistent and reproducible teleoperation testing under tight latency constraints. As per the srsRAN documentation, the E2 interface is still under development with limited features, and the srsUE component has constraints in 5G SA mode, including limited bandwidth support and lack of handover functionality. Additionally, stability issues such as UE disconnections after a few minutes have been observed.

We emphasize the following to show the novelty of our O-RAN integration:

- The system incorporates the full integration of a Near-RT RIC with srsRAN (CU/DU split) using E2 interfaces and two functional xApps, an extended version of the KPIMON xApp, and a custom xApp within a robotic teleoperation context. To the best of our knowledge, such a complete and reproducible integration for real-time robotic control is not yet widely explored in the literature.
- The extended KPIMON xApp, based on the OAIC implementation [18], was used for real-time sensing of network telemetry and RIC indication messages. Specifically, the KPIMON SM was configured to periodically monitor the PDCP bytes transmitted to and from the srsUE haptic interface and the robotic arm, with a 1-second event trigger interval, while both UEs maintained continuous connectivity to the eNB. This xApp enabled network state awareness at the RIC level without requiring deep packet inspection.
- The custom xApp, also based on OAIC templates [19], was designed to monitor MCS alongside BLER using a lightweight machine learning algorithm. The ML logic uses simple rule-based classification to detect degradation patterns in MCS/BLER trends without invoking complex or computationally heavy inference models. Consequently, no significant ML model complexity is necessary in this use case, as long as BLER and latency remain within defined tolerable limits. The purpose of this monitoring framework is to establish the groundwork for future closed-loop control extensions, including dynamic MCS reconfiguration, predictive congestion management, or adaptive scheduling, all of which can be built incrementally atop the current xApp framework.
- Overall, this setup introduces an application-aware O-RAN telemetry pipeline, where robotic QoS constraints (e.g., low BLER and sub-50 ms latency) are linked to RAN behaviour, a first step toward cross-layer control in open RAN-enabled cyber-physical systems. This platform can serve as a stepping stone for future 5G-based robotic and XR applications.

Strengths of the Paper: *The authors successfully deployed a complete end-to-end 4G LTE system using open-source srsRAN and USRP hardware, applying it to a teleoperation use case involving a robotic arm and a haptic controller.*

Response: Authors are grateful to the reviewer for the positive feedback and critical review of the manuscript.

Areas for improvement 1: Section III – O-Cloud Terminology: The statement "The O-Cloud comprises two primary units..." is misleading. If the O-CU and O-DU are deployed on bare-metal servers, this does not constitute an O-Cloud. The term O-Cloud is appropriate only when referring to a virtualized or cloud-native deployment environment.

Response: We thank the reviewer for pointing out the misuse of terminology in Section III regarding the O-Cloud. We acknowledge that, according to O-RAN Alliance definitions, the term O-Cloud refers specifically to a cloud-native or virtualized infrastructure platform that hosts O-RAN functions such as O-CU, O-DU, and RIC components. This typically involves containerized or virtual machine deployments orchestrated by platforms like Kubernetes or OpenStack.

In our current testbed, the O-CU and O-DU are deployed on bare-metal servers without virtualization, using srsRAN on Ubuntu hosts. Therefore, referring to this setup as an O-Cloud is technically incorrect. We will revise the manuscript to replace the term "O-Cloud" with "physical testbed" or "bare-metal server deployment" where appropriate. Please to our highlighted response in blue font colour in the revised manuscript at section III (Data Transmission Model) – page 6.

Areas for improvement 2: *Formulae (1)-(7): The relevance of formulae (1) through (7) is unclear. Ultimately, the paper uses formula (8), which is derived from 3GPP or O-RAN specifications. However, the citation "[24] O-RAN website" is vague and unhelpful. The authors should cite the exact technical document (e.g., O-RAN TIFG E2E Test R0003-v06.00, Section 6.1) to support the formula.*

Response: We thank the reviewer for this important observation. We acknowledge that the connection between formulae (1) through (7) and the final throughput expression in formula (8) was not clearly explained in the original draft. We agree that "[24] O-RAN website" is too vague. We will revise the citation to refer specifically to the relevant O-RAN technical document. Please refer to our below highlighted response in blue font colour in the revised manuscript at section III (Data Transmission Model) – page 6/7 and we highlight it below as well:

To provide a foundational understanding of how wireless channel conditions influence achievable data rates, we begin with a sequence of theoretical expressions, outlined in equations (1) through (7). These equations illustrate the progressive refinement from idealized channel capacity models—based on Shannon’s theory and SINR—to more practical representations that incorporate system-specific parameters such as modulation order Q_m , coding rate R_c , and allocated bandwidth in terms of Physical Resource Blocks (PRBs).

While these expressions are not directly used in the performance analysis, they serve to conceptually frame how radio conditions (e.g., SINR) are mapped into modulation and coding configurations, and ultimately, throughput. These intermediate steps also help contextualize the relationship between channel estimation quality and application-layer performance in our robotic teleoperation system.

Building upon this theoretical framework, we adopt the standardized physical layer throughput expression defined in the O-RAN Technical Specification R0003-v06.00 [27], Section 6.1, as shown in equation (8). This expression integrates all relevant system parameters—including PRB allocation, modulation scheme, coding rate, control overhead, and OFDM symbol timing—into a practical and measurable rate computation for both downlink and uplink scenarios.

Areas for improvement 3: *Theoretical vs. Experimental Throughput: There is a significant discrepancy between the theoretical throughput (as derived from formulas (8)–(10)) and the actual throughput observed in Section IV. The authors should clarify the purpose of presenting the theoretical throughput and provide explanations for the observed differences.*

Response: We thank the reviewer for this important observation. We added new subsection {Theoretical vs. Experimental Throughput} as a concluding paragraph within our existing performance discussion. Please refer to our below highlighted response in blue font colour in the revised manuscript at section IV subsection A (Theoretical vs. Experimental Throughput) – page 9 and we highlight it below as well:

While equations (8)– (10) provide a theoretical model for physical-layer throughput based on modulation order Q_m , coding rate R_c , PRB allocation, and OFDM symbol timing, they represent idealized upper bounds. These expressions assume full PRB utilization, perfect link conditions, negligible overhead, and no scheduler inefficiencies.

In practice, our experimental results—collected via iperf3 at the application layer—demonstrate significantly lower throughput values due to various system-level and hardware constraints.

For example, using 100 PRBs, 64-QAM ($Q_m = 6$), and $R_c \approx 0.9258$, the theoretical downlink throughput reaches approximately 43.05 Mbps. However, as shown in Fig.5, the observed downlink throughput for both the haptic controller and robotic arm ranges between 9.4–10.5 Mbps. Uplink throughput is even lower, limited to 0.3–0.4 Mbps due to lower MCS values and protocol overhead

This discrepancy is expected and primarily results from:

- MAC-layer inefficiencies and partial PRB utilization, please refer to Table II,
- Control and RRC signaling overheads,
- Buffering delays and software radio latency in the srsRAN stack,
- Retransmissions and imperfect SINR conditions during indoor experiments,
- Periodic PDCP byte transmission patterns, confirmed in Fig. 8

This section helps clarify the practical gap between spec-based models and observed results, while reinforcing the relevance of empirical validation.

To summarize the comparison, we provide the following table:

Table: Comparison of Theoretical and Measured Throughput

Metric	Theoretical Value	Measured Value	Notes
MCS Index (DL)	28 (64-QAM)	18–27	Adaptive variation based on SINR
PRBs Used	100	Partial (CLI-confirmed)	Scheduling gaps observed in logs
Downlink Throughput	74.34 Mbps	9.4–10.5 Mbps	Measured via iperf3
Uplink MCS Index	N/A	13–21	Limits UL performance
Uplink Throughput	N/A	0.3–0.4 Mbps	Limited by UL channel and lower MCS
BLER	~0% (ideal)	6–12%	Increases with higher MCS (see Fig. 7)
PDCP Byte Trend	N/A	Periodic burst (Fig. 8)	Matches KPIMON trigger every 1s

Areas for improvement 4: *DLUL Ratio in Formula (10): The use of DLUL_ratio = 0.543 is incorrect for LTE Band 7 in FDD mode, where uplink and downlink operate on different center frequencies. This ratio is typically used in TDD mode. For FDD, the DLUL_ratio should be 1.*

Response: We thank the reviewer for pointing this out and agree that the DLUL_ratio term is typically relevant for TDD systems, where time-domain slot allocation is split between downlink and uplink symbols. In contrast, FDD systems such as LTE Band 7 use separate frequency bands for DL and UL, and in strict theoretical modeling, the DLUL_ratio should indeed be set to 1 to reflect full allocation. In our example of formula (10), we followed the throughput example provided in O-RAN TIFG E2E Test Specification (R0003-v06.00), page 50, which uses a value of DLUL_ratio = 0.543 even though the duplexing mode is not explicitly specified. We now realize this value likely reflects a TDD scenario or simplified frame utilization model for specific test conditions.

Please refer to our below highlighted response in blue font colour in the revised manuscript at section III (Data Model Transmission) – page 7 and we highlight it below as well:

Numerical example: Let us consider $BW = 20$ MHz, $N_{PRB}^{BW} = 100$, $OH = 0.15$, $MCS=28$, the modulation 64 QAM, $Q_m = 6$, $R_{code} = 0.9258$ and $DLUL_{ratio} = 1$

$$R_{th} = \left(6 \times 0.9258 \times \frac{100 \times 12}{\frac{10^{-3}}{14}} (1 - 0.15) \times 1 \right)$$

$$= 74.34 \text{ Mbps}$$

Equation (10), which calculates the physical layer data rate, is adapted from the O-RAN TIFG E2E Test Specification \cite{oran_r0003}. While the original document includes an example using a DLUL_ratio of 0.543—likely corresponding to a TDD system—we revise this to reflect our actual deployment using LTE Band 7 in FDD mode.

In FDD, uplink and downlink transmissions occur simultaneously on separate frequency bands. Therefore, all symbols allocated for the downlink are considered usable, and the DLUL_ratio term should be set to 1.

Areas for improvement 5: *Figure 6 – MCS and Bitrate Discrepancy: In Figure 6 (top), only MCS DL = 27 is shown, which appears to be an error. Even if MCS DL varies from 20 to 27, the corresponding DL bitrate fluctuates around 10 Mbps, which contradicts the observations described in Section IV. Also, combining UL and DL observations into a single sentence is misleading and oversimplifies the system behavior.*

Response: We thank the reviewer for this precise observation and agree that Figure 6 (Figure 5 in the revised manuscript) could be better annotated and described to clarify the variation in MCS DL and its corresponding impact on bitrate.

Upon rechecking the source trace logs used to generate Figure 6 (Figure 5 in the revised manuscript) (see uploaded file: *Robotic arm user*), we confirm that **MCS DL does vary over time**—ranging from 27 down to 20. For example, in the first few seconds of the trace, the MCS DL consistently holds at 27, then gradually drops to 26, 25, and eventually reaches 20–21 during lower signal conditions. This trend is correctly reflected in the full dataset but was **underrepresented in the early visual snapshot** of Fig. 5, which only included a small window (~10 seconds). We will update Fig. 6 to display a **longer time window**, showing the actual variation in MCS DL values more clearly.

Regarding the DL bitrate behavior: While the **instantaneous bitrate at MCS = 27 hovers around 10 Mbps**, we observe clear **fluctuations** when MCS drops to lower values (20–21), where **bitrate degrades to ~0.3 Mbps**, consistent with the explanation provided in Section IV. We will improve the figure caption and text in the manuscript to explicitly state that **DL bitrate is proportional to MCS** within observable SINR ranges, and that **brief stable phases at MCS = 27** lead to temporary throughput plateaus.

We also acknowledge the reviewer’s point about combining DL and UL results into a single sentence. We will revise this section to **separate the DL and UL discussions** for clarity and avoid misleading generalizations.

Please refer to our below highlighted response in blue font colour in the revised manuscript at section IV (Experimental Results) – page 8 and we highlight it below as well:

Fig.5 illustrates the relationship between Modulation and Coding Scheme (MCS) values and the corresponding bitrate for both downlink (DL) and uplink (UL) directions for the robotic arm user. While MCS DL reaches a peak value of 27 during high-SINR conditions, the log traces confirm that it periodically drops to values between 20 and 21 under interference or signal degradation.

The DL bitrate remains near 10 Mbps when MCS DL = 27, but significantly declines (as low as 0.3 Mbps) when MCS values decrease. This validates the expected dependency between MCS and achievable throughput. We have revised the plotted time window of Fig.5 to better reflect this variation.

To avoid oversimplification, we have also separated DL and UL discussions. The UL MCS remains within a narrower range (13–21), resulting in consistently lower throughput due to uplink path loss and lower scheduling priority. This asymmetry highlights the importance of analyzing DL and UL behaviors independently in real-time teleoperation scenarios.

Fig.5: Time evolution of MCS and bitrate for the robotic arm user. (Top) Downlink: MCS DL fluctuates from 27 to 20, corresponding to variations in DL bitrate from 10 Mbps down to 2 Mbps. (Bottom) Uplink: MCS UL ranges between 13–21, resulting in UL bitrate between 0.4 and 0.8 Mbps. The figure illustrates the dynamic adaptation behavior over a 60-second test window.

Areas for improvement 6: *Custom xApp Functionality:* The role and operation of the custom xApp require more detailed explanation. The statement that it can "identify network interference and dynamically modify the MCS to counteract its effects" is vague and raises several important questions:

o Did the authors modify the MCS selection mechanism in the eNB? According to 3GPP standards, the eNB determines MCS based on CQI feedback from the UE, considering CSI-RS, SINR, BLER, and HARQ performance.

o If the xApp overrides this process, how does that affect UE behavior and overall system compatibility?

- *Lack of Closed-Loop Control:* Overall, the purpose of the custom xApp remains unclear. Without a closed-loop system—where KPIMon monitors performance and the xApp applies control commands based on optimization within the Near-RT RIC—the use of a custom xApp lacks justification.

Response: We sincerely thank the reviewer for raising this important point regarding the scope and functionality of the custom xApp.

To clarify: in our current work, we employed **two xApps** from the OpenAI Cellular (OAIC) repository:

- The **custom xApp** for lightweight monitoring of MCS and BLER statistics \cite{xapp_oaic}, and
- The **KPIMON xApp**, which reports RIC metrics such as PDCP bytes, CQI, and other UE-level statistics \cite{kpimon_oaic}.

We confirm that **no modifications were made** to the xApps' logic or to the srsENB MCS selection mechanism. The MCS continues to be selected by the eNB based on CQI feedback from the UE, in full compliance with the 3GPP standard mechanism involving CSI-RS measurements, SINR, and HARQ feedback. As such, the **custom xApp does not override, intercept, or inject control commands** into the RAN. The xApp's role in our testbed was solely to **observe trends in MCS and BLER**, log these at the Near-RT RIC, and visualize their impact under changing network conditions.

We acknowledge the reviewer's observation that without closed-loop actuation, the xApp's functionality is currently **limited to monitoring**. However, our objective was to demonstrate the **integration of a working telemetry path** between the RAN and RIC using open interfaces and publicly available xApp templates. This serves as a **foundation or stepping stone** for future work where closed-loop control—such as dynamic MCS tuning, slice reconfiguration, or latency-aware scheduling—can be added on top.

The key insight is that even without modifying the xApp or inserting new ML models, the system achieved acceptable **latency (<50 ms)** and **BLER thresholds**, showing that real-time robotic communication can be monitored through the Near-RT RIC under O-RAN principles. Our goal was to **evaluate the testbed stability and telemetry integration**, not yet to demonstrate policy-based RAN control.

We have updated the manuscript to clarify our below highlighted response in blue font colour in the revised manuscript at section II A (O-RAN system for real-time robotics) – page 4 and we highlight it below as well:

In this work, two open-source xApps from the OpenAI Cellular (OAIC) project were integrated into our testbed and deployed in the Near-RT RIC environment:

- KPIMON xApp — used to monitor real-time network statistics such as PDCP bytes, CQI reports, and UE identifiers via E2SM-KPM Service Model. It was configured with a 1-second periodic trigger to log performance data for both the haptic controller and the robotic arm UEs.

- Custom xApp — used to monitor and visualize Modulation and Coding Scheme (MCS) and Block Error Rate (BLER) trends over time. It runs a lightweight Python process that subscribes to telemetry and displays variations in link-layer parameters.

Importantly, no modifications were made to either xApp. The MCS selection mechanism within the srsRAN eNB was left unmodified, adhering to standard 3GPP procedures where MCS is selected based on CQI feedback derived from SINR and HARQ performance. The custom xApp does not override or inject control commands into the RAN, nor does it interfere with the eNB's decision-making logic.

The primary role of these xApps in our study was to establish a working telemetry and analytics pipeline, enabling real-time monitoring of RAN behavior in a robotic teleoperation scenario. This architecture serves as a foundation for future closed-loop control, where

KPIMON data could be used to trigger xApp-driven optimization policies (e.g., adaptive MCS tuning or latency-aware scheduling).

Despite not implementing closed-loop control, the system achieved acceptable real-time performance — maintaining downlink latency below 50 ms and BLER within tolerable limits — thus validating the feasibility of using lightweight, standards-aligned xApps for robotic use cases under O-RAN architecture.

Finally, we would like to thank the Reviewer for constructive comments that helped us to considerably improve the technical and presentation quality of our manuscript.

Author's Response Letter:

Manuscript ID: manuscript COMMS-ENG-25-0245

Paper Title: Development of Open RAN for Real-time Teleoperation

Authors: Saber Hassouna, Jaspreet Kaur , Burak Kizilkaya, Jalil Kazim, Shuja Ansari, Arzad Alam Kherani, Brijesh Lall, Qammer H. Abbasi, and Muhammad Ali Imran

The authors would like to thank the Editor and the Reviewers for their time and valuable comments. We have revised our manuscript and have addressed the specific comments of the reviewers and the editor below in this response letter. We have highlighted the changes in the blue font color in the revised version of the manuscript and the reasonings for these changes are given in the responses below.

We hope that the concerns of the Editor and the Reviewers have been appropriately addressed in the revised version and the paper is accepted for publication in this form.

EDITOR'S COMMENTS:

General Comments: *Thank you for your letter asking us to reconsider our decision on your manuscript entitled "Development of Open RAN for Real-time Teleoperation". After careful assessment of your response we have decided that we would be willing to consider a revised version of your manuscript along the lines you have described, provided that nothing similar has been accepted at Communications Engineering or published elsewhere by the time the revised manuscript is received. If the manuscript is suitably revised, we will go back to the same reviewers for their comments.*

Response: We would like to thank the Editor for the careful review of our manuscript. We have made extensive changes and thoroughly revised our manuscript, according to the useful feedback received by the Editor and the Reviewers. The changes are duly highlighted in blue font color in the revised version of the manuscript.

Finally, we would like to thank the Editor for constructive comments that helped us to considerably improve the technical and presentation quality of our manuscript.

REVIEWER 1'S COMMENTS:

General Comments: *This paper presents an O-RAN testbed design with robotic end devices, i.e., a haptic controller and a robotic arm. The use case is teleoperation for dental inspections. The system is designed using open-source libraries and software. Experiments show transmission stability in both the Uplink and the Downlink. In terms of latency, the system could operate near-real-time, showing the potential to be used in teleoperations. Some questions and suggestions about this work are:*

Response: Authors are grateful to the reviewer for the positive feedback and critical review of the manuscript.

Comments 1: *Would you consider making your implementation open-source? It would be a valid contribution to the community.*

Response 1: Authors are thankful to the reviewer for the valuable comment. In response to this valuable comment, we would like to clarify and confirm that when our work is accepted for publication, we will make it open source to the public.

Comment 2: *Fig. 4 is not informative and hard to read. Please consider changing it to a table instead.*

Response 2: Authors are thankful to the reviewer for the valuable comment. In response to this valuable comment, please refer to our below highlighted response in blue font colour in the revised manuscript at section IV (Experimental results) – page 7 and we highlight it below as well:

In this study, we present the metrics obtained from the default trace in the command line interface (CLI) tool for srsRAN. Tables III and IV show the trace of the srsRAN eNB where metrics are provided for the DL and UL respectively.

TABLE III: The trace of srsRAN eNB DL metrics

rat	rnti	cqi	ri	mcs	brate	ok	nok	(%)
lte	4f	10	0	27	9.8M	291	6	2%
lte	4f	10	0	27	11M	323	10	3%
lte	4f	10	0	27	10.0M	308	8	2%
lte	4f	10	0	26	11M	338	17	4%
lte	4f	10	0	26	10M	335	16	4%
lte	4f	10	0	26	11M	346	16	4%
lte	4f	10	0	26	10M	335	8	2%
lte	4f	10	0	27	10M	323	10	3%
lte	4f	10	0	27	11M	319	7	2%
lte	4f	10	0	27	11M	325	7	2%

TABLE IV: The trace of srsRAN eNB UL metrics

pusch	pucch	phr	mcs	brate	ok	nok	(%)	bsr
33.2	23.7	31	21	287k	65	0	0%	0.0
31.1	23.9	31	22	328k	63	0	0%	0.0
20.4	23.9	31	19	342k	73	36	33%	0.0
18.6	24.7	31	19	323k	67	43	39%	0.0
18.5	24.7	31	17	296k	75	36	32%	0.0
17.4	24.6	31	16	322k	75	29	27%	0.0
17.5	24.3	31	15	305k	77	24	23%	0.0
17.4	24.2	31	14	275k	73	18	20%	0.0
16.6	24.3	31	13	316k	64	15	18%	0.0
18.5	24.6	31	14	312k	80	13	13%	0.0

Comment 3: *On page 6, it mentioned that "This implementation requires deploying a Kubernetes-based environment with Docker containers, incorporating machine learning models for interference detection". Could you elaborate on the machine learning models? Was that some automated MSC adaptation methods? Would you consider improving the MSC adaptation algorithms?*

Response 3: Authors are thankful to the reviewer for the valuable comment. In response to this valuable comment, please refer to our below highlighted response in blue font colour in the revised manuscript at section IV (Experimental results) – bottom of page 7 and we highlight it below as well:

The custom xApp leverages a Kubernetes-based deployment using Docker containers and integrates machine learning (ML) models for real-time interference classification. We used the implementation [22] which adopts lightweight ML classifiers (such as decision trees or logistic regression) for distinguishing interference types based on features extracted from RIC telemetry data (e.g., RSRP, SINR, and throughput).

We would like to emphasize that this custom xApp, with its simple ML model, was successfully integrated and tested in our fully deployed end-to-end 4G LTE system. This system is built using open-source srsRAN and USRP hardware, and supports a teleoperation use case — where low-latency and reliable communication are critical. In this practical setup, we found that the use of advanced or complex ML algorithms, such as reinforcement learning or context-aware optimization, was not necessary, as long as system performance metrics such as block error rate (BLER) and latency remained within acceptable thresholds. The xApp sends a command to start using adaptive MCS, the trace will show the MCS values changing to around 13-21, according to the signal quality.

Finally, we would like to thank the Reviewer for constructive comments that helped us to considerably improve the technical and presentation quality of our manuscript.

REVIEWER 2'S COMMENTS:

General Comments: *This is a testbed design paper where the authors are implementing the O-RAN testbed for teleoperation through two UEs. Overall the idea is very relevant and implementation. There have been a few papers which has proposed similar ideas which he authors have also acknowledged.*

I think this paper has a potential to make impact and influence the field. However, I am a bit concerned about the aspect of it. There are a few concerns which are not clear from this version:

Response: Authors are grateful to the reviewer for the positive feedback and critical review of the manuscript.

Comment 1: *What is the overall framework? A representative figure of each block would have been helpful. What type of data is being transmitted during the experiment?*

Response 1: We thank the reviewer for this insightful comment. please refer to our below highlighted response in blue font colour in the revised manuscript at section II (O-RAN Network Design and Implementation) – bottom of page 3 and we highlight it below as well:

Addressing the request regarding the overall framework and the representative figures:

The proposed system architecture is shown in Fig. 1 of the manuscript, which provides a high-level overview of the Open RAN-based framework integrating robotic control and haptic feedback via a fully deployed 4G LTE testbed. This includes the near-real-time RIC, the custom xApp, and the end-to-end connectivity between the operator-side haptic interface and the teleoperator robotic arm. Fig. 2 further clarifies the experimental testbed setup, including the servers, USRPs, mobile dongles, and UE hardware and Fig. 3 illustrates the physical deployment of the base station setup and connected devices.

Addressing the type of transmitted data:

The system transmits real-time control commands, such as position and velocity vectors from the haptic interface to the robot, and force feedback signals in the reverse direction. These are encapsulated in standard IP packets over LTE and routed through the srsRAN-based EPC. In addition, the iperf3 was used to generate UDP traffic in some test phases to characterize the LTE throughput under controlled load conditions, as described in the OAI Cellular documentation [20].

Comment 2: *The use case specific analysis, rather than randomly transmitting traffic through iperf, is it possible to generate a few haptic sequence for experiment setting, and evaluate them. A more organized experiment set would be very helpful for potentially establishing such idea.*

Response 2:

We thank the reviewer for this insightful comment. please refer to our below highlighted response in blue font colour in the revised manuscript at section IV (experimental results) – beginning of page 7 and we highlight it below as well:

In this study, we present the metrics obtained from the default trace in the command line interface (CLI) tool for srsRAN. Tables III and IV show the trace of the srsRAN eNB where metrics are provided for the DL and UL respectively. In our current work, the iperf3 tool was primarily used to generate and analyze baseline LTE throughput under different radio configurations in a controlled and repeatable manner. This allowed us to validate the end-to-end system performance of the open-source srsRAN stack and USRP hardware under various MCS, while keeping the application-layer traffic consistent and quantifiable. We acknowledge that iperf3-generated traffic does not capture the full dynamics of real-time teleoperation data.

Beyond the iperf3 traffic, our experimental framework was also used with real haptic command sequences and feedback signals between the operator and the robot. The haptic stream includes time-series data such as position vectors and force feedback sampled at 100–1000 Hz, encapsulated in LTE packets and exchanged over the deployed Open RAN network. These sequences were streamed using a real robotic teleoperation interface, similar in spirit to prior work such as in [2].

Comment 3: *Not clear how the data transmission model is relevant - why we need intermediate Rayleigh channels, how about experiment on real channel in various settings? After all this is a real setup.*

Response 3:

We thank the reviewer for this insightful comment. please refer to our below highlighted response in blue font colour in the revised manuscript at section III (Data Transmission Model) – bottom of page 6 and we highlight it below as well:

The inclusion of the analytical data transmission model in our paper was intended to complement the empirical results obtained from the physical testbed. Specifically, the model serves as a tool to understand the upper bounds of the achievable throughput based on radio parameters and physical layer characteristics, and to explain how factors such as MCS, resource blocks, and control overhead affect link capacity.

While the setup is indeed based on a real end-to-end O-RAN-compliant system using srsRAN and USRP hardware, we used the model to relate channel quality (SINR) to achievable rate through its effect on the modulation order Q_m and coding rate R_c . As mentioned in the O-RAN Alliance specifications [27], the SINR does not explicitly appear in the physical layer throughput equation (8), but it indirectly determines the modulation and coding scheme (MCS) thus influencing both Q_m and R_c . Thus, our analysis is grounded in empirical measurements, and the data model serves as a complementary tool to understand how radio configuration parameters (e.g., PRB allocation, MCS) impact achievable throughput. Consequently, no synthetic channel model (e.g., Rayleigh fading) was used in the performance results. All results reflect actual physical wireless channel conditions within our test environment.

Finally, we would like to thank the Reviewer for constructive comments that helped us to considerably improve the technical and presentation quality of our manuscript.

REVIEWER 3'S COMMENTS:

1- Summary of the Paper: *This paper presents an end-to-end 4G LTE open-source system based on srsRAN, integrated with the O-RAN Near-RT RIC and two xApps: KPIMon and a custom xApp displaying MCS and BLER metrics. The system is evaluated in a teleoperation scenario involving a robotic arm and a haptic controller.*

However, the work does not demonstrate any novel utilization of O-RAN capabilities or significant use of O-RAN open interfaces tailored for teleoperation applications.

Additionally, deploying low-latency robotic control over 4G LTE is not state-of-the-art, particularly considering the availability of 5G technology. With the same USRP X310 hardware, the authors could have deployed the srsRAN 5G stack and tested with a 5G dongle or srsUE stack to better align with modern teleoperation requirements.

Furthermore, there is no compelling justification for deploying the custom xApp solely to monitor MCS and BLER, especially in the absence of any concrete control actions taken by the xApp to improve teleoperation performance (e.g., latency reduction or reliability enhancement).

Response: We appreciate the reviewer's thoughtful feedback and the opportunity to clarify the contributions of our work. Please refer to our below highlighted response in blue font colour in the revised manuscript at section I (Introduction) – page 3 and we highlight it below as well:

Our primary objective was to demonstrate the feasibility of deploying an open, standards-aligned, end-to-end O-RAN system using open-source platforms (srsRAN and OpenAirInterface components) for real-time robotic teleoperation. While it is true that the use of 5G would provide lower latency and higher reliability, our motivation for selecting a 4G LTE stack was grounded in practical accessibility and current deployment maturity.

The open-source srsRAN 5G stack, while evolving, currently exhibits limitations in stability and full-stack integration, particularly at the UE and RIC interface layers, required for consistent and reproducible teleoperation testing under tight latency constraints. As per the srsRAN documentation, the E2 interface is still under development with limited features, and the srsUE component has constraints in 5G SA mode, including limited bandwidth support and lack of handover functionality. Additionally, stability issues such as UE disconnections after a few minutes have been observed.

We emphasize the following to show the novelty of our O-RAN integration:

- The system incorporates the full integration of a Near-RT RIC with srsRAN (CU/DU split) using E2 interfaces and two functional xApps, an extended version of the KPIMON xApp, and a custom xApp within a robotic teleoperation context. To the best of our knowledge, such a complete and reproducible integration for real-time robotic control is not yet widely explored in the literature.
- The extended KPIMON xApp, based on the OAIC implementation [18], was used for real-time sensing of network telemetry and RIC indication messages. Specifically, the KPIMON SM was configured to periodically monitor the PDCP bytes transmitted to and from the srsUE haptic interface and the robotic arm, with a 1-second event trigger

interval, while both UEs maintained continuous connectivity to the eNB. This xApp enabled network state awareness at the RIC level without requiring deep packet inspection.

- The custom xApp, also based on OAIC templates [19], was designed to monitor MCS alongside BLER using a lightweight machine learning algorithm. The ML logic uses simple rule-based classification to detect degradation patterns in MCS/BLER trends without invoking complex or computationally heavy inference models. Consequently, no significant ML model complexity is necessary in this use case, as long as BLER and latency remain within defined tolerable limits. The purpose of this monitoring framework is to establish the groundwork for future closed-loop control extensions, including dynamic MCS reconfiguration, predictive congestion management, or adaptive scheduling, all of which can be built incrementally atop the current xApp framework.
- Overall, this setup introduces an application-aware O-RAN telemetry pipeline, where robotic QoS constraints (e.g., low BLER and sub-50 ms latency) are linked to RAN behaviour, a first step toward cross-layer control in open RAN-enabled cyber-physical systems. This platform can serve as a stepping stone for future 5G-based robotic and XR applications.

2- Strengths of the Paper: *The authors successfully deployed a complete end-to-end 4G LTE system using open-source srsRAN and USRP hardware, applying it to a teleoperation use case involving a robotic arm and a haptic controller.*

Response: Authors are grateful to the reviewer for the positive feedback and critical review of the manuscript.

3- Areas for Improvement:

Areas for improvement 1: Section III – O-Cloud Terminology: The statement "The O-Cloud comprises two primary units..." is misleading. If the O-CU and O-DU are deployed on bare-metal servers, this does not constitute an O-Cloud. The term O-Cloud is appropriate only when referring to a virtualized or cloud-native deployment environment.

Response: We thank the reviewer for pointing out the misuse of terminology in Section III regarding the O-Cloud. We acknowledge that, according to O-RAN Alliance definitions, the term O-Cloud refers specifically to a cloud-native or virtualized infrastructure platform that hosts O-RAN functions such as O-CU, O-DU, and RIC components. This typically involves containerized or virtual machine deployments orchestrated by platforms like Kubernetes or OpenStack.

In our current testbed, the O-CU and O-DU are deployed on bare-metal servers without virtualization, using srsRAN on Ubuntu hosts. Therefore, referring to this setup as an O-Cloud is technically incorrect. We will revise the manuscript to replace the term "O-Cloud" with "physical testbed" or "bare-metal server deployment" where appropriate. Please to our highlighted response in blue font colour in the revised manuscript at section III (Data Transmission Model) – page 4.

Areas for improvement 2: *Formulae (1)-(7): The relevance of formulae (1) through (7) is unclear. Ultimately, the paper uses formula (8), which is derived from 3GPP or O-RAN specifications. However, the citation "[24] O-RAN website" is vague and unhelpful. The authors should cite the exact technical document (e.g., O-RAN TIFG E2E Test R0003-v06.00, Section 6.1) to support the formula.*

Response: We thank the reviewer for this important observation. We acknowledge that the connection between formulae (1) through (7) and the final throughput expression in formula (8) was not clearly explained in the original draft. We agree that “[24] O-RAN website” is too vague. We will revise the citation to refer specifically to the relevant O-RAN technical document. Please refer to our below highlighted response in blue font colour in the revised manuscript at section III (Data Transmission Model) – beginning of page 6 and we highlight it below as well:

To provide a foundational understanding of how wireless channel conditions influence achievable data rates, we begin with a sequence of theoretical expressions, outlined in equations (1) through (7). These equations illustrate the progressive refinement from idealized channel capacity models—based on Shannon’s theory and SINR—to more practical representations that incorporate system-specific parameters such as modulation order Q_m , coding rate R_c , and allocated bandwidth in terms of Physical Resource Blocks (PRBs).

While these expressions are not directly used in the performance analysis, they serve to conceptually frame how radio conditions (e.g., SINR) are mapped into modulation and coding configurations, and ultimately, throughput. These intermediate steps also help contextualize the relationship between channel estimation quality and application-layer performance in our robotic teleoperation system.

Building upon this theoretical framework, we adopt the standardized physical layer throughput expression defined in the O-RAN Technical Specification R0003-v06.00 [27], Section 6.1, as shown in equation (8). This expression integrates all relevant system parameters—including PRB allocation, modulation scheme, coding rate, control overhead, and OFDM symbol timing—into a practical and measurable rate computation for both downlink and uplink scenarios.

Areas for improvement 3: *Theoretical vs. Experimental Throughput: There is a significant discrepancy between the theoretical throughput (as derived from formulas (8)–(10)) and the actual throughput observed in Section IV. The authors should clarify the purpose of presenting the theoretical throughput and provide explanations for the observed differences.*

Response: We thank the reviewer for this important observation. We added new subsection {Theoretical vs. Experimental Throughput} as a concluding paragraph within our existing performance discussion. Please refer to our below highlighted response in blue font colour in the revised manuscript at section IV subsection A (Theoretical vs. Experimental Throughput) – bottom of page 8 and we highlight it below as well:

While equations (8)– (10) provide a theoretical model for physical-layer throughput based on modulation order Q_m , coding rate R_c , PRB allocation, and OFDM symbol timing, they represent idealized upper bounds. These expressions assume full PRB utilization, perfect link conditions, negligible overhead, and no scheduler inefficiencies.

In practice, our experimental results—collected via iperf3 at the application layer—demonstrate significantly lower throughput values due to various system-level and hardware constraints.

For example, using 100 PRBs, 64-QAM ($Q_m = 6$), and $R_c \approx 0.9258$, the theoretical downlink throughput reaches approximately 74.34 Mbps. However, as shown in Fig.5, the observed downlink throughput for both the haptic controller and robotic arm ranges between 9.4–10.5 Mbps. Uplink throughput is even lower, limited to 0.3–0.4 Mbps due to lower MCS values and protocol overhead

This discrepancy is expected and primarily results from:

- MAC-layer inefficiencies and partial PRB utilization, please refer to Table II,
- Control and RRC signaling overheads,
- Buffering delays and software radio latency in the srsRAN stack,
- Retransmissions and imperfect SINR conditions during indoor experiments,
- Periodic PDCP byte transmission patterns, confirmed in Fig. 8

This section helps clarify the practical gap between spec-based models and observed results, while reinforcing the relevance of empirical validation.

To summarize the comparison, we provide the following table:

TABLE V: Comparison of Theoretical and Measured Throughput

Metric	Theoretical	Measured	Notes
MCS Index	28 (64-QAM)	18–27 (DL)	Adaptively lowered based on SINR variations
PRBs Used	100	Partial	Periodic gaps in scheduling
DL Throughput	74.34 Mbps	9.4–10.5 Mbps	Measured with iperf3. Transmission initiated with: <code>sudo iperf3 -c 162.16.0.1 -p 5006 -i 1 -t 180 -R -b 10M</code> , targeting 10 Mbps for benchmarking against theoretical maximum
UL MCS Index	N/A	13–21	Limited by uplink link quality
UL Throughput	N/A	0.3–0.4 Mbps	Affected by conservative MCS and traffic profile
BLER	~0% (ideal)	6–12%	See Fig. 7; increases with higher MCS
PDCP Byte Trend	N/A	Periodic bursts	Matches KPIMON event triggers every 1s

Areas for improvement 4: *DLUL Ratio in Formula (10): The use of $DLUL_ratio = 0.543$ is incorrect for LTE Band 7 in FDD mode, where uplink and downlink operate on different center frequencies. This ratio is typically used in TDD mode. For FDD, the $DLUL_ratio$ should be 1.*

Response: We thank the reviewer for pointing this out and agree that the DLUL_ratio term is typically relevant for TDD systems, where time-domain slot allocation is split between downlink and uplink symbols. In contrast, FDD systems such as LTE Band 7 use separate frequency bands for DL and UL, and in strict theoretical modeling, the DLUL_ratio should indeed be set to 1 to reflect full allocation. In our example of formula (10), we followed the throughput example provided in O-RAN TIFG E2E Test Specification (R0003-v06.00), page 50, which uses a value of DLUL_ratio = 0.543 even though the duplexing mode is not explicitly specified. We now realize this value likely reflects a TDD scenario or simplified frame utilization model for specific test conditions.

Please refer to our below highlighted response in blue font colour in the revised manuscript at section III (Data Model Transmission) – page 7 and we highlight it below as well:

Numerical example: Let us consider $BW = 20$ MHz, $N_{PRB}^{BW} = 100$, $OH = 0.15$, $MCS=28$, the modulation 64 QAM, $Q_m = 6$, $R_{code} = 0.9258$ and $DLUL_{ratio} = 1$

$$R_{th} = \left(6 \times 0.9258 \times \frac{100 \times 12}{\frac{10^{-3}}{14}} (1 - 0.15) \times 1 \right)$$

$$= 74.34 \text{ Mbps}$$

Equation (10), which calculates the physical layer data rate, is adapted from the O-RAN TIFG E2E Test Specification \cite{oran_r0003}. While the original document includes an example using a DLUL_ratio of 0.543—likely corresponding to a TDD system—we revise this to reflect our actual deployment using LTE Band 7 in FDD mode.

In FDD, uplink and downlink transmissions occur simultaneously on separate frequency bands. Therefore, all symbols allocated for the downlink are considered usable, and the DLUL_ratio term should be set to 1.

Areas for improvement 5: *Figure 6 – MCS and Bitrate Discrepancy: In Figure 6 (top), only MCS DL = 27 is shown, which appears to be an error. Even if MCS DL varies from 20 to 27, the corresponding DL bitrate fluctuates around 10 Mbps, which contradicts the observations described in Section IV. Also, combining UL and DL observations into a single sentence is misleading and oversimplifies the system behavior.*

Response: We thank the reviewer for this precise observation and agree that Figure 6 (Figure 5 in the revised manuscript) could be better annotated and described to clarify the variation in MCS DL and its corresponding impact on bitrate.

Upon rechecking the source trace logs used to generate Figure 6 (Figure 5 in the revised manuscript), we confirm that MCS DL does vary over time—ranging from 27 down to 20. For example, in the first few seconds of the trace, the MCS DL consistently holds at 27, then gradually drops to 26, 25, and eventually reaches 20–21 during lower signal conditions. This trend is correctly reflected in the full dataset but was **underrepresented in the early visual**

snapshot of Fig. 5, which only included a small window (~10 seconds). We will update Fig. 5 to display a **longer time window**, showing the actual variation in MCS DL values more clearly.

Regarding the DL bitrate behavior: While the **instantaneous bitrate at MCS = 27 hovers around 10 Mbps**, we observe clear **fluctuations** when MCS drops to lower values (20–21), where **bitrate degrades to ~0.3 Mbps**, consistent with the explanation provided in Section IV. We will improve the figure caption and text in the manuscript to explicitly state that **DL bitrate is proportional to MCS** within observable SINR ranges, and that **brief stable phases at MCS = 27** lead to temporary throughput plateaus.

We also acknowledge the reviewer’s point about combining DL and UL results into a single sentence. We will revise this section to **separate the DL and UL discussions** for clarity and avoid misleading generalizations.

Please refer to our below highlighted response in blue font colour in the revised manuscript at section IV (Experimental Results) – bottom of page 7 and we highlight it below as well:

Fig.5 illustrates the relationship between Modulation and Coding Scheme (MCS) values and the corresponding bitrate for both downlink (DL) and uplink (UL) directions for the robotic arm user. While MCS DL reaches a peak value of 27 during high-SINR conditions, the log traces confirm that it periodically drops to values between 20 and 21 under interference or signal degradation.

The DL bitrate remains near 10 Mbps when MCS DL = 27, but significantly declines (as low as 0.3 Mbps) when MCS values decrease. This validates the expected dependency between MCS and achievable throughput. We have revised the plotted time window of Fig.5 to better reflect this variation.

To avoid oversimplification, we have also separated DL and UL discussions. The UL MCS remains within a narrower range (13–21), resulting in consistently lower throughput due to uplink path loss and lower scheduling priority. This asymmetry highlights the importance of analyzing DL and UL behaviors independently in real-time teleoperation scenarios.

Fig.5: Time evolution of MCS and bitrate for the robotic arm user. (Top) Downlink: MCS DL fluctuates from 27 to 20, corresponding to variations in DL bitrate from 10 Mbps down to 2 Mbps. (Bottom) Uplink: MCS UL ranges between 13–21, resulting in UL bitrate between 0.4 and 0.8 Mbps. The figure illustrates the dynamic adaptation behavior over a 60-second test window.

Areas for improvement 6: *Custom xApp Functionality: The role and operation of the custom xApp require more detailed explanation. The statement that it can "identify network interference and dynamically modify the MCS to counteract its effects" is vague and raises several important questions:*

o Did the authors modify the MCS selection mechanism in the eNB? According to 3GPP standards, the eNB determines MCS based on CQI feedback from the UE, considering CSI-RS, SINR, BLER, and HARQ performance.

o If the xApp overrides this process, how does that affect UE behavior and overall system compatibility?

• Lack of Closed-Loop Control: Overall, the purpose of the custom xApp remains unclear. Without a closed-loop system—where KPIMon monitors performance and the xApp applies control commands based on optimization within the Near-RT RIC—the use of a custom xApp lacks justification.

Response: We sincerely thank the reviewer for raising this important point regarding the scope and functionality of the custom xApp.

To clarify: in our current work, we employed **two xApps** from the OpenAI Cellular (OAIC) repository:

- The **custom xApp** for lightweight monitoring of MCS and BLER statistics [22], and
- The **KPIMON xApp**, which reports RIC metrics such as PDCP bytes, CQI, and other UE-level statistics [21].

We confirm that **no modifications were made** to the xApps' logic or to the srsENB MCS selection mechanism. The MCS continues to be selected by the eNB based on CQI feedback from the UE, in full compliance with the 3GPP standard mechanism involving CSI-RS measurements, SINR, and HARQ feedback. As such, the **custom xApp does not override, intercept, or inject control commands** into the RAN. The xApp's role in our testbed was solely to **observe trends in MCS and BLER**, log these at the Near-RT RIC, and visualize their impact under changing network conditions.

We acknowledge the reviewer's observation that without closed-loop actuation, the xApp's functionality is currently **limited to monitoring**. However, our objective was to demonstrate the **integration of a working telemetry path** between the RAN and RIC using open interfaces and publicly available xApp templates. This serves as a **foundation or stepping stone** for future work where closed-loop control—such as dynamic MCS tuning, slice reconfiguration, or latency-aware scheduling—can be added on top.

The key insight is that even without modifying the xApp or inserting new ML models, the system achieved acceptable **latency (<50 ms)** and **BLER thresholds**, showing that real-time robotic communication can be monitored through the Near-RT RIC under O-RAN principles. Our goal was to **evaluate the testbed stability and telemetry integration**, not yet to demonstrate policy-based RAN control.

We have updated the manuscript to clarify our below highlighted response in blue font colour in the revised manuscript at section II A (O-RAN system for real-time robotics) – bottom page 3 and we highlight it below as well:

In this work, two open-source xApps from the OpenAI Cellular (OAIC) project were integrated into our testbed and deployed in the Near-RT RIC environment:

- KPIMON xApp — used to monitor real-time network statistics such as PDCP bytes, CQI reports, and UE identifiers via E2SM-KPM Service Model. It was configured with a 1-second periodic trigger to log performance data for both the haptic controller and the robotic arm UEs.

- Custom xApp — used to monitor and visualize Modulation and Coding Scheme (MCS) and Block Error Rate (BLER) trends over time. It runs a lightweight Python process that subscribes to telemetry and displays variations in link-layer parameters.

Importantly, no modifications were made to either xApp. The MCS selection mechanism within the srsRAN eNB was left unmodified, adhering to standard 3GPP procedures where MCS is selected based on CQI feedback derived from SINR and HARQ performance. The custom xApp does not override or inject control commands into the RAN, nor does it interfere with the eNB's decision-making logic.

The primary role of these xApps in our study was to establish a working telemetry and analytics pipeline, enabling real-time monitoring of RAN behavior in a robotic teleoperation scenario. This architecture serves as a foundation for future closed-loop control, where KPIMON data could be used to trigger xApp-driven optimization policies (e.g., adaptive MCS tuning or latency-aware scheduling).

Despite not implementing closed-loop control, the system achieved acceptable real-time performance — maintaining downlink latency below 50 ms and BLER within tolerable limits — thus validating the feasibility of using lightweight, standards-aligned xApps for robotic use cases under O-RAN architecture.

Finally, we would like to thank the Reviewer for constructive comments that helped us to considerably improve the technical and presentation quality of our manuscript.

Author's Response Letter:

Manuscript ID: manuscript COMMS-ENG-25-0245B

Paper Title: Development of Open RAN for Real-time Teleoperation

Authors: Saber Hassouna, Jaspreet Kaur , Burak Kizilkaya, Jalil Kazim, Shuja Ansari, Arzad Alam Kherani, Brijesh Lall, Qammer H. Abbasi, and Muhammad Ali Imran

The authors would like to thank the Editor and the Reviewers for their time and valuable comments. We have revised our manuscript and have addressed the specific comments of the reviewers and the editor below in this response letter. We have highlighted the changes in the blue font color in the revised version of the manuscript and the reasonings for these changes are given in the responses below.

We hope that the concerns of the Editor and the Reviewers have been appropriately addressed in the revised version and the paper is accepted for publication in this form.

EDITOR'S COMMENTS:

General Comments: *Thank you for your letter asking us to reconsider our decision on your manuscript entitled "Development of Open RAN for Real-time Teleoperation". After careful assessment of your response we have decided that we would be willing to consider a revised version of your manuscript along the lines you have described, provided that nothing similar has been accepted at Communications Engineering or published elsewhere by the time the revised manuscript is received. If the manuscript is suitably revised, we will go back to the same reviewers for their comments.*

Response: We would like to thank the Editor for the careful review of our manuscript. We have made extensive changes and thoroughly revised our manuscript, according to the useful feedback received by the Editor and the Reviewers. The changes are duly highlighted in blue font color in the revised version of the manuscript.

Finally, we would like to thank the Editor for constructive comments that helped us to considerably improve the technical and presentation quality of our manuscript.

REVIEWER 1'S COMMENTS:

General Comments: *My concerns are addressed. Recommend acceptance.*

Response: Authors are grateful to the reviewer for the positive feedback and critical review of the manuscript.

REVIEWER 2'S COMMENTS:

General Comments: *Just do a final typo and grammatical error checking - Typo: 'block error rate (Bler)' - > 'BLER' at page 7. Full form is not mentioned at the first occurrence of BLER - at abstract also in the introduction. First time it is expanded is in section II, page 3. Please perform a thorough typo checking.*

Response: Authors are grateful to the reviewer for the positive feedback and critical review of the manuscript. All the typo errors have been corrected.

REVIEWER 3'S COMMENTS:

General Comments: *The authors have addressed most of the critical points raised in my previous review. I appreciate their efforts in improving the manuscript.*

Response: Authors are grateful to the reviewer for the positive feedback and critical review of the manuscript.

Questions and Clarifications:

Comment 1: *srsRAN 5G Stack Limitations: The authors claim that the srsRAN 5G stack exhibits limitations in terms of stability and full-stack integration. Could the authors elaborate on the specific aspects or components where these limitations are observed?*

Response 1: We thank the reviewer for requesting clarification. In the revised manuscript (**Section I, paragraph 8**), we now explicitly highlight several limitations observed during our practical experimentation.

- **Stability:** Based on internal testing, we encountered repeated **UE disconnections after 2–5 minutes** during downlink traffic generation in standalone (SA) mode. This was particularly evident when operating without external GPSDO synchronization. This aligns with issues raised by the srsRAN community (see below), where internal timing drift or HARQ malfunctions caused Radio Link Failures (RLF).
 - GitHub Issue #533: UE disconnects after a few seconds
 - GitHub Issue #1227: Segmentation faults on 5G SA mode
- **Lack of end-to-end support for advanced 5G features:** While srsRAN 5G supports gNB-5GC communication in SA mode, the implementation lacks:
 - **Handover support**
 - **Full bandwidth support beyond 20 MHz**, despite 5G NR specifications allowing up to 100 MHz per carrier.
 - **Robust support for E2 interface in 5G stack:** As of srsRAN 21.10, **native E2 agent integration in 5G** was not fully released or documented. While the 4G (srsLTE) stack has an OAIC-modified E2 agent (see Reference [17]), the

5G branch requires additional manual adaptation, as noted in community discussions and project forks.

These limitations affected our ability to build a stable, feature-rich 5G SA testbed. Therefore, our final system leveraged both 4G (with E2 support) and 5G stacks (for standalone evaluation), depending on our use case.

References to Response 1:

1. GitHub Issue #533 – https://github.com/srsran/srsRAN_Project/issues/533
2. GitHub Issue #1227 – https://github.com/srsran/srsRAN_Project/issues/1227
3. OAIC srsRAN + E2 documentation (Reference [17]) – https://openaicellular.github.io/oaic/srsRAN_installation.html

Comment 2: *RIC Integration Support: According to the srsRAN website and GitHub repository, RIC integration is natively supported in the 5G stack. Could the authors clarify what additional integration or modification was required in their implementation?*

Response 2: We thank the reviewer for raising this important point. We would like to clarify that **no additional modifications or custom integration were required in our implementation.** We used the **existing OAIC-supported srsRAN 4G (srsLTE) stack**, which includes an integrated E2 agent as described in:

- OAIC documentation: https://openaicellular.github.io/oaic/srsRAN_installation.html
- OAIC's GitHub repository: <https://github.com/openaicellular/srsRAN-e2>

While it is true that the **srsRAN 5G stack has recently added experimental support for E2**, its publicly available feature set remains limited. As of the latest release notes [1], the open-source 5G version supports only:

- **Two service models:** E2SM-KPM and E2SM-RC
- **No O1 management support**, which is only available in the enterprise edition and listed as a future roadmap item [1, 2]

We therefore deliberately chose the mature 4G stack for the following reasons:

- **Stability:** In our tests using UHD 4.7 and srsRAN 5G v21.10, we observed intermittent UE disconnections and unreproducible behavior.
- **Functional completeness:** The 4G stack provided stable KPM metric extraction and reliable integration with the Near-RT RIC via E2 termination.
- **Reproducibility:** Using 4G ensured low-latency, deterministic behavior suitable for teleoperation data capture, while still aligning with the O-RAN architecture.

References to response 2:

1. srsRAN 5G Release Notes – https://docs.srsran.com/projects/project/en/latest/general/source/5_release_notes.html
2. Sthamer, P. (2024). *Evaluation of srsRAN's 5G Support for O-RAN* – https://www.dn.th-koeln.de/wp-content/uploads/2024/06/Bachelorarbeit-Pascal_Sthamer-final.pdf

Comment 3: *CU/DU Split in srsRAN 4G Stack: The concept of CU/DU disaggregation is commonly associated with 5G. In my understanding, srsRAN's 4G stack (i.e., srsLTE) compiles the eNB as a single monolithic instance, without explicit CU/DU split. Could the authors provide a credible reference or documentation to support the claim that CU/DU disaggregation is available in the 4G stack? If this is a novel contribution or workaround, more details would be helpful.*

Response 3: We thank the reviewer for this accurate observation. We fully agree that the **srsRAN 4G stack (srsLTE)** compiles the eNB as a **monolithic implementation** and does not support protocol-level CU/DU disaggregation as defined in 3GPP 5G architecture.

In our implementation, we **did not attempt to implement or emulate a CU/DU split** in the 4G stack. Rather, the control and user-plane components operate within a single executable, and all E2-related functions are integrated at the eNB level. While srsLTE's internal architecture (e.g., multithreaded processing of PDCP, RRC, MAC) loosely reflects modularity, this is **not equivalent to disaggregated network functions**, nor do we suggest that it constitutes true CU/DU separation.

We have now:

- **Removed any implication** that the 4G stack supports CU/DU disaggregation.
- Clearly stated in **Section I – Paragraph 9** that our architecture preserves the monolithic nature of the 4G eNB, with no additional separation beyond what is inherently present in srsLTE's software design.
- Referenced [11] in our manuscript, which surveys several experimental 4G testbeds and confirms the absence of native CU/DU splits in such setups.

Our O-RAN testbed was designed to be **modular at the orchestration level**, allowing future integration of disaggregated gNB components once srsRAN's 5G support matures, but no such disaggregation was performed in this 4G deployment.

Reference to Response 3:

- Ref [11]: Testbed survey confirming that CU/DU disaggregation is not natively supported in 4G experimental setups and is typically introduced only in custom 5G architectures.

Comment 4: *srsUE Limitations in 5G SA Mode: The authors mention that srsUE has constraints in 5G SA mode, such as limited bandwidth support and lack of handover functionality.*

- *Why does srsUE need to distinguish between SA and NSA mode, given its role as a user equipment?*
- *Could the authors clarify how bandwidth limitations manifest in 5G compared to 4G for srsUE?*
- *Please provide references or documentation to support the claim about lack of handover support in srsUE.*

Response 4: We appreciate the reviewer’s detailed questions on srsUE capabilities in 5G SA. Our clarification is three-fold: (i) why srsUE must distinguish SA vs. NSA, (ii) how bandwidth limits manifest for srsUE in 5G versus LTE, and (iii) the current handover status.

(i) Why srsUE must distinguish SA vs. NSA.

In NSA the UE is anchored on LTE (E-UTRA) and is dual-connected to NR, whereas in SA the UE is anchored solely on NR and a 5G Core. This architectural split drives different RRC/NAS procedures, measurement reporting and bearer setup, so a UE implementation necessarily exposes mode-specific configuration and control-plane handling. This is reflected both in 3GPP stage-2 specs (e.g., TS 37.340 for multi-connectivity/EN-DC and TS 38.300 for NR overall description) and in srsUE’s separate configuration sections for LTE and NR modes in SA tutorials. In short, srsUE needs to “know” the mode because the signalling stacks and procedures differ across SA and NSA by design.

(ii) Bandwidth/SCS limits of srsUE in 5G SA vs. LTE.

The current public documentation for srsRAN’s “srsRAN gNB with srsUE” tutorial lists explicit 5G SA limitations for srsUE: 15 kHz SCS only (FDD bands), bandwidth limited to 5/10/15/20 MHz, and no handover [1]. These are given in the Limitations box on that page. Release notes and public slides also confirm that the project recently fixed/validated 5/10/15/20 MHz operation in 5G SA, reinforcing the above bandwidth envelope. By contrast, LTE srsUE has long-standing support across the standard LTE bandwidths 1.4–20 MHz, so the 5G SA path is currently no wider than LTE and additionally constrained by 15 kHz SCS, which precludes common 30 kHz FR1 TDD deployments (e.g., n78) in typical lab setups [2]. This is the practical reason we retained the LTE UE path for repeatable experiments.

[1] https://docs.srsran.com/projects/4g/en/latest/general/source/2_release_notes.html

[2] https://docs.srsran.com/projects/4g/en/latest/usermanuals/source/srsue/source/1_ue_intro.html

(iii) Handover support status.

The srsRAN gNB gained intra-gNB handover capability in Project 24.04 [3]; however, the srsUE 5G (SA) client does not support handover at this time, as stated in the official tutorial’s Limitations and reiterated by maintainers in multiple discussions. Consequently, even where the RAN side can initiate an NR mobility event, the open srsUE cannot complete a 5G HO procedure, which again informed our choice to avoid claiming 5G SA mobility results

[3] <https://docs.srsran.com/projects/project/en/latest/tutorials/source/handover/source/index.html>

Reference to Response 4:

[1] https://docs.srsran.com/projects/4g/en/latest/general/source/2_release_notes.html

[2] https://docs.srsran.com/projects/4g/en/latest/usermanuals/source/srsue/source/1_ue_intro.html

[3] <https://docs.srsran.com/projects/project/en/latest/tutorials/source/handover/source/index.html>

Comment 5: *Stability Issues in srsRAN 5G Stack: The authors mention observing stability issues such as UE disconnections after a few minutes. Could they provide more concrete evidence or references for this observation?*

Based on my own experience with the srsRAN 5G stack (as of 2024), it was relatively stable, and recent versions have shown further improvements. This claim seems surprising and warrants further clarification.

Response 5: We appreciate the reviewer's comment and fully recognize that the **srsRAN 5G stack has matured significantly** in recent versions.

However, our experiments were conducted using **srsRAN version 21.10**, integrated with **UHD 4.7** and **USRP X310** hardware. Despite applying recommended practices such as GPSDO-based clock synchronization and precise sampling rates, we sometimes encountered **unstable behavior** in standalone (SA) mode. These included **UE disconnections shortly after registration** and **irregular failures during NAS session setup**, which disrupted our ability to capture consistent performance traces.

These behaviors were **consistent with known issues** reported in the srsRAN community:

- GitHub Issue #1227: "UE disconnected after registration in SA mode"
https://github.com/srsran/srsRAN_Project/issues/1227
- GitHub Issue #533: "Unstable behavior with Open5GS + SA mode"
https://github.com/srsran/srsRAN_Project/issues/533

Additionally, the official srsRAN documentation srsRAN gNB with srsUE — srsRAN Project documentation explicitly notes that:

"Standalone mode currently has limitations due to incomplete NAS implementation. Handover is not supported."

These factors made it difficult to conduct repeatable, low-latency evaluations required for our target use case (i.e., remote teleoperation with near real-time responsiveness). For this reason, we opted to proceed with the **stable and well-documented 4G stack**, where our setup proved robust and reproducible.

Comment 6: *Choice of xApp over rApp: Why did the authors choose to implement the monitoring component as an xApp? Are there technical or architectural reasons that make xApp preferable over an rApp in this context? I do not see value of monitoring sub-second performance metrics with xApp.*

Response 6: We thank the reviewer for this insightful question.

Our choice of an xApp is architecturally aligned with O-RAN's timing model and with the teleoperation QoS we target. By specification and widely accepted practice, the Near-RT RIC operates in the ~10 ms–1 s window and hosts xApps that subscribe to and act upon RAN telemetry via E2; the Non-RT RIC hosts rApps on > 1 s to minutes time scales and exchanges policies/models over A1 (not streaming telemetry). For a 1 s telemetry cadence (E2SM-KPM periodic indications), the Near-RT RIC/xApp is therefore the correct placement, while an rApp would be intentionally slower and focused on policy orchestration/training rather than sub-second observability [1][2].

Concretely, our xApp subscribes via E2SM-KPM with a 1 s periodic Indication to collect PDCP byte counts, CQI/MCS-related KPIs, and UE identifiers directly from the E2 node. This follows the intended E2→Near-RT data path; rApps do not subscribe over E2 and instead interact through A1 to provide policy guidance and models to the Near-RT RIC. Using an rApp for this function would add avoidable latency and remove direct access to the indication stream we analyse [3].

Regarding “value”: teleoperation is sensitive to short-lived BLER/MCS excursions that can degrade haptics well within a 1s window. Sub-second monitoring at the Near-RT layer provides (i) timely alarm/telemetry for operators and (ii) a drop-in path to closed-loop control (e.g., scheduling hints or traffic shaping) in future work without relocating the application. In contrast, an rApp would be appropriate for long-horizon optimization (e.g., policy updates, offline training) but not for the immediate observability we need in this study.

To avoid any ambiguity, we have updated the manuscript to state explicitly that the xApp in this work is monitoring-only (no override of 3GPP MCS logic); it processes 1 s E2SM-KPM indications in the Near-RT RIC, while policy-level actions via A1/rApps are left to future extensions. Please, refer to section 2A, Paragraph 2.

[1] <https://rimedolabs.com/blog/o-ran-near-real-time-ric>

[2] Municio, G. Garcia-Aviles, A. Garcia-Saavedra and X. Costa-Pérez, "O-RAN: Analysis of Latency-Critical Interfaces and Overview of Time Sensitive Networking Solutions," in *IEEE Communications Standards Magazine*, vol. 7, no. 3, pp. 82-89, September 2023

[3] <https://docs.srsran.com/projects/project/en/latest/tutorials/source/near-rt-ric/source/index.html>

Comment 7: *ML Logic in Custom xApp: Regarding the ML-based xApp:*

- *What type of ML task is being used—classification, regression, or something else?*

- *How was the model trained, and on what data?*

- *What is the specific objective of using ML here?*

If the xApp only identifies degradation patterns in metrics such as MCS or BLER using simple rules, it is unclear why an ML model is needed. Further justification is required.

Response 7: We appreciate the request for clarity. To avoid overstating, we have revised the manuscript to remove the “ML-based” phrasing for this release. The component is now described as a rules-based heuristic detector that operates on E2SM-KPM indications at 1s cadence. It does not override 3GPP MCS selection and does not require any model training. Its sole purpose in this paper is timely, near-RT monitoring with well-defined trigger conditions that can be used to raise operator alerts and to scaffold future closed-loop control.

What type of ML task is used?

None in this release. We renamed the module to a rules-based heuristic detector. It emits a binary degradation event when concurrent conditions are met.

How was the model trained; on what data?

No model training is performed. Thresholds are set a priori (see below). For transparency, we added the exact thresholds and a small sensitivity analysis in the appendix.

Specific objective of “ML” here?

The objective is early detection of short-lived degradations relevant to teleoperation (e.g., brief BLER/MCS excursions) within the Near-RT (≤ 1 s) window, to (i) notify the operator and (ii) provide a drop-in telemetry-to-action hook for a future closed-loop xApp. In this monitoring-only scope, simple rules are adequate and preferable to a trained model.

Reference to Response 7:

https://openaircellular.github.io/oaic/xapp_python.html

Comment 8: *Throughput Discrepancy Explanation: The explanation for the discrepancy between theoretical and measured throughput is vague and lacks supporting evidence. The reasons provided are too generic and implicitly suggest that srsRAN 4G’s implementation is subpar, which needs to be substantiated with data or references.*

Response 8: We thank the reviewer for this important observation. To improve the technical clarity and evidence supporting our throughput analysis, we would like to clarify that equation (10) in the revised manuscript is the ideal throughput calculations. The 74.34 Mb/s comes from the spec-based upper bound (full 100 PRB utilization, 64-QAM with high code rate, FDD, nominal overhead), following the O-RAN TIFG throughput expression (eqs. 8–10). This is not

a prediction for our run; it is an analytic ceiling. Our downlink was intentionally capped at ~10 Mb/s by the offered load. In all DL tests we used iperf3 ... -R -b 10M, i.e., reverse mode at a fixed 10 Mb/s target. Consequently, no DL measurement can exceed 10 Mb/s by design, which explains the plateau around 9.4–10.5 Mb/s in Fig. 4/Tab. III. We now state this explicitly in Section IV and in the caption/notes (the command is already shown in Table V).

Comment 9: *Figure 9 – Latency vs. Packet Loss: The correlation between latency and packet loss in Figure 9 is unclear. For example, in the time range 0–5s, packet loss is high (up to 3.5%) while latency remains low and stable (~110ms). Conversely, when packet loss drops below 1%, latency becomes more volatile. The authors are encouraged to re-examine this relationship or provide an explanation for this discrepancy.*

Response 9: Thank you for pointing this out. We agree that the prior text did not explicitly explain why latency spikes and loss spikes may not be time-aligned. We have revised the manuscript to (i) define the latency metric we plotted, (ii) describe the two operating regimes observed in the trace, and (iii) add supporting layer-2 counters that make the behavior clear.

Metric definition & alignment (now stated in the text/caption).

We clarify that “Latency (ms)” in Fig. 9 is application-layer RTT measured at 1 s granularity, while “Packet Loss (%)” is computed over the same 1 s bins. Because the two series are sampled rather than per-packet and are affected by different buffering/retransmission processes, contemporaneous peaks need not coincide.

Why latency peaks can occur without simultaneous loss.

In LTE, HARQ (PHY/MAC) and RLC-AM retransmissions can queue and delay packets while still delivering them successfully, producing latency spikes with little or no loss. HARQ rounds are specified at MAC (3GPP TS 36.321), and RLC-AM recovery, including t-Reordering/t-Poll Retransmit timers, is specified in RLC (3GPP TS 36.322). When these mechanisms are active (e.g., transient radio impairments), end-to-end delay can increase noticeably without producing drops at the application layer.

Why loss can appear with modest latency.

Conversely, when discard timers/policies at RLC-AM/PDCP expire (e.g., if recovery exceeds a configured bound), packets may be dropped promptly, yielding *loss spikes* with only a small contemporaneous latency increase. PDCP/RLC timer behavior (e.g., PDCP reordering/discard, RLC-AM STATUS timing) is standardized (TS 36.323/TS 36.322); operational discussions widely note that a tighter PDCP discard window can surface as loss even when mean latency stays moderate.

We now note this in Section IV, 11th paragraph.

Finally, we would like to thank the Reviewer for constructive comments that helped us to considerably improve the technical and presentation quality of our manuscript.

Author's Response Letter:

Manuscript ID: manuscript COMMS-ENG-25-0245C-Z

Paper Title: Development of Open RAN for Real-time Teleoperation

Authors: Saber Hassouna, Jaspreet Kaur, Burak Kizilkaya, Jalil Kazim, Shuja Ansari, Arzad Alam Kherani, Brijesh Lall, Qammer H. Abbasi, and Muhammad Ali Imran

The authors would like to thank the Editor and the Reviewers for their time and valuable comments. We have revised our manuscript and have addressed the specific comments of the reviewers and the editor below in this response letter.

We hope that the concerns of the Editor and the Reviewers have been appropriately addressed in the revised version and the paper is accepted for publication in this form.

EDITOR'S COMMENTS:

General Comments: *While the reviewer has approved the current version, during the editorial assessment we have noticed that some of the less critical points have only been partially addressed and we would like the authors to consider the following to improve the standing of the study.*

Regarding the claimed limitations of srsRAN 5G Stack, we kindly request to be explicit and up front about the necessity and reasons for the choice of the specific software versions used in the paper. In addition, we kindly request to clearly discuss the version limitations and comment on how a different software may change the output.

Response: We thank the editor for highlighting the need to be more explicit about our software version choices and the associated limitations. In response, we have revised the manuscript (Section I, Paragraphs 9,10,11, pages 2 and 3) to clearly justify the selected versions and explicitly discuss their constraints. We highlight the response below as well.

We conducted all experiments using **srsRAN-LTE v21.10**, combined with the **OAIC E2 agent** and the "Cherry v2.0" Near-RT RIC container. This version was selected because it represented the most recent stable LTE release available when data collection began (May 2025), and it supported a robust 1-second E2SM-KPM telemetry stream—critical for enabling our sub-second control loop in robotic teleoperation.

We evaluated **srsRAN-5G v24.04**, but encountered several blockers that prevented reliable experimentation:

- (1) the open-source branch at the time supported only E2SM-KPM and a single RC Style 2, without O1 interface support
- (2) community issue #1227 (UE deregistration in SA mode) consistently reproduced in our lab
- (3) the SA UE exhibited missing counters and intermittent segmentation faults, which disrupted closed-loop telemetry collection.

Given these issues, we chose to proceed with the mature LTE stack for this study. However, we note that **srsRAN-5G v24.10**, released after our experiments, has introduced additional features, including CU/DU split, E2SM-RC Style 3, and stability improvements. Once support for SA handover, wider bandwidths (e.g., 30 kHz SCS), and a complete O1 interface is fully stabilized in the open branch, our modular testbed can be readily migrated to a native 5G stack without architectural changes. Our software design already anticipates this forward compatibility.

Finally, we would like to thank the Editor for constructive comments that helped us to considerably improve the technical and presentation quality of our manuscript.

REVIEWER 3'S COMMENTS:

General Comments: *The authors have addressed most of the critical points raised in my previous review. I appreciate their efforts in improving the manuscript.*

Response: Authors are grateful to the reviewer for the positive feedback and critical review of the manuscript.

Finally, we would like to thank the Reviewer for constructive comments that helped us to considerably improve the technical and presentation quality of our manuscript.